# SEMF: Supervised Expectation-Maximization Framework for Predicting Intervals

## Abstract

This work introduces the Supervised Expectation-Maximization Framework (SEMF), a versatile and model-agnostic approach for generating prediction intervals in datasets with complete or missing data. SEMF extends the Expectation-Maximization algorithm, traditionally used in unsupervised learning, to a supervised context, leveraging latent variable modeling for uncertainty estimation. Extensive empirical evaluations across 11 tabular datasets show that SEMF often achieves narrower normalized prediction intervals and higher coverage rates than traditional quantile regression methods. Furthermore, SEMF can be integrated with machine learning models like gradient-boosted trees and neural networks, highlighting its practical applicability. The results indicate that SEMF enhances uncertainty quantification, particularly in scenarios with complete data.

## 1 Introduction

In the evolving field of machine learning (ML), the quest for models able to predict outcomes while quantifying the uncertainty of their predictions is critical. The ability to estimate prediction uncertainty is particularly vital in high-stakes domains such as healthcare (Dusenberry et al., 2020), finance (Wisniewski et al., 2020), and autonomous systems (Tang et al., 2022), where prediction-based decisions have important consequences. Traditional approaches have primarily focused on point estimates, with little to no insight into prediction reliability. This limitation underscores the need for frameworks that can generate both precise point predictions and robust prediction intervals. Such intervals provide a range within which the true outcome is expected to lie with a fixed probability, offering a finer understanding of prediction uncertainty. This need has spurred research into methodologies that extend beyond point estimation to include uncertainty quantification, thereby enabling more informed decision-making in applications reliant on predictive modeling (Ghahramani, 2015).

In this paper, we introduce the Supervised Expectation-Maximization Framework (SEMF) based on the Expectation-Maximization (EM) algorithm (Dempster et al., 1977). Traditionally recognized as a clustering technique, EM is used for supervised learning in SEMF, allowing for both point estimates and prediction intervals using any ML model (model-agnostic). SEMF generates representations for latent or missing modalities, which can be relevant for incomplete data and holds potential for multi-modal data applications, though multi-modal settings are left for future exploration. This paper details the methodology behind the framework and proposes a training algorithm based on Monte Carlo (MC) sampling, also used in variational inference for Variational Auto-Encoders (VAEs) (David M. Blei & McAuliffe, 2017a; Kingma & Welling, 2014). SEMF differs from prominent supervised EM approaches such as Ghahramani & Jordan (1993), which focus on point prediction using Gaussian Mixture Models (GMMs). Additionally, our method operates in a frequentist paradigm, directly maximizing the likelihood function through iterative EM steps without integrating over posterior distributions. Although SEMF can be extended to a Bayesian framework as its likelihood component, this extension lies beyond the scope of this paper.

The remainder of this paper is organized as follows: Section 2 details the theory and the methodology of SEMF. Section 3 reviews related works in latent representation learning, uncertainty estimation, and handling of missing data. Section 4 describes the experimental setup, including datasets and evaluation metrics. Section 5 discusses the results, demonstrating the efficacy of SEMF. Lastly, Section 6 concludes the paper, and Section 7 outlines the limitations and potential research directions.

## 2 METHOD

This section presents the founding principles of SEMF from its parameters, training, and inference procedure with, at its core, the EM algorithm. This algorithm, first introduced by Dempster et al. (1977), is an unsupervised method for handling latent variables and incomplete data. Invented to maximize the model likelihood, it builds a sequence of parameters that guarantee an increase in the log-likelihood (Wu, 1983) by iterating between the Expectation (E) and the Maximization (M) steps. In the E-step, one computes

$$Q(p|p') = \mathbb{E}_{Z \sim p'(z|x)} [\log p(x, Z)] = \int \log p(x, z) p'(z|x) dz, \tag{1}$$

where $p'$ stands for the current estimates, $\log p(x, z)$ is the log-likelihood of the complete observation $(x, z)$, and $z$ is a latent variable. The M-step maximizes this $Q$-function: $p' \leftarrow \arg\max_p Q(p|p')$. The sequence is repeated until convergence.

### 2.1 PROBLEM SCENARIO

Let $x = (x_1, x_2, \ldots, x_K)$ denote $K$ inputs and the output be $y$. For simplicity, we limit $y$ to be numerical, although it could be categorical without loss of generality. Component $x_k$ is a source: a modality, a single or group of variables, or an unstructured input such as an image or text. For clarity, we limit to $K = 2$, where $x_1$ and $x_2$ are single variables. We assume that only $x_1$ may contain missing values, either at random or partially at random.

Let $p(y|x)$ be the density function of the outcome given the inputs (the fact that $y$ is continuous can be easily relaxed). A founding assumption, in the spirit of VAE, is that $p(y|x)$ decomposes into $p(y|x) = \int p(y|z) p(z_1|x_1) p(z_2|x_2) dz_1 dz_2$, where $z = (z_1, z_2)$ are unobserved latent variables. We assume that $p(y|z, x) = p(y|z)$, that is, $z$ contains all the information of $x$ about $y$, and that $p(z|x) = p(z_1|x_1) p(z_2|x_2)$, that is, there is one latent variable per source. These are independent conditionally on their corresponding source. Finally, if $x_1$ is missing, then $p(y|x_2) = \int p(y|z) p(z_1|x_1) p(x_1|x_2) p(z_2|x_2) dx_1 dz_1 dz_2$. The contribution to the log-likelihood of a complete observation $(y, z, x)$ is $\log p(y, z|x) = \log p(y|z) + \log p(z|x)$. In the E-step, we compute

$$\int \log p(y, z|x) p'(z|y, x) dz = \int \log p(y|z) p'(z|y, x) dz + \int \log p(z|x) p'(z|y, x) dz. \tag{2}$$

where $p'$ is our current estimate. Eq. 2 can be estimated by MC sampling. Since sampling from $p'(z|y, x)$ can be inefficient, we rather rely on the decomposition $p'(z|y, x) = p'(y|z) p'(z|x) / p'(y|x)$. Thus, we sample $z_r$ from $p'(z|x)$, $r = 1, \ldots, R$, and, setting $w_r = p'(y|z_r) / \sum_t p'(y|z_t)$, approximate the right-hand side term of Eq. 2

$$\int \log p(y, z|x) p'(z|y, x) dz \approx \sum_{r=1}^{R} \{\log p(y|z_r) + \log p(z_r|x)\} w_r. \tag{3}$$

If $x_1$ is missing, a similar development leads to

$$\int \log p(y, z, x_1|x_2) p'(z, x_1|y) dz \approx \sum_{r=1}^{R} \{\log p(y|z_r) + \log p(z_r|x) + \log p(x_{1,r}|x_2)\} w_r, \tag{4}$$

where $x_{1,r}$ and $z_r$ are respectively sampled from $p'(x_1|x_2)$ and $p'(z|x_{1,r}, x_2)$.

### 2.2 OBJECTIVE FUNCTION

Adapting Eq. 3 and Eq. 4 for the observed data $\{(y_i, x_i)\}_{i=1}^{N}$, the overall loss function, $\mathcal{L}$, is

$$\mathcal{L}(\phi, \theta, \xi) = -\sum_{i=1}^{N} \sum_{r=1}^{R} \{\log p_\phi(z_{i,r}|x_{i,r}) + \log p_\theta(y_i|z_{i,r}) + \mathbb{1}_{\{i \in I_m\}} \log p_\xi(x_{1,i,r}|x_{2,i})\} w_{i,r}, \tag{5}$$

where $I_m$ is the set of those $i$'s such that $x_{1,i}$ is missing. The models of $p(y|z)$, $p(z|x)$, and $p(x_1|x_2)$ inherit parameters $\theta$, $\phi$, and $\xi$, respectively. Also, $x_{1,i,r}$ is sampled from $p_{\xi'}(x_1|x_{2,i})$, if $x_{1,i}$ is missing,

and $z_{1,i,r}$ and $z_{2,i,r}$ are sampled from $p_{\phi'}(z_1|x_{1,i,r})$ and $p_{\phi'}(z_2|x_{2,i})$. Furthermore, for compactness of notation, $x_{i,r}$ is $(x_{1,i}, x_{2,i})$ if $x_{1,i}$ is observed, and $(x_{1,i,r}, x_{2,i})$ if $x_{1,i,r}$ is missing. Finally, the weights are

$$w_{i,r} = \frac{p_{\theta'}(y_i|z_{i,r})}{\sum_{t=1}^{R} p_{\theta'}(y_i|z_{i,t})}. \tag{6}$$

Eq. 5 shows that $\mathcal{L}$ is a sum of losses associated with the encoder model, $p_\phi$, for each source, the decoder model, $p_\theta$, from the latent variables to the output, and, if applicable, the model handling missing data, $p_\xi$. At each M-step, $\mathcal{L}$ is minimized with respect to $\theta$, $\phi$, and $\xi$. Then, $\theta'$, $\phi'$, and $\xi'$ are updated, as well as the weights and the sampling. Then the process is iterated until convergence.

### 2.2.1 Example: $\mathcal{L}(\phi, \theta, \xi)$ under normality

Similar to Kingma & Welling (2014), we develop further $\mathcal{L}$ under normality assumptions for the encoder, $p_\phi$, and the decoder, $p_\theta$. These simple cases are illustrative, though any other distributions could be adopted, including non-continuous or non-numerical outcomes.

**Encoder $p_\phi(z|x)$.** Let $m_k$ be the length of the latent variable $z_k$, $k = 1, 2$. We assume a normal model for $Z_k$ given $X_k = x_k$,

$$Z_k|X_k = x_k \sim \mathcal{N}_{m_k}(g_{\phi_k}(x_k), \sigma_k^2 J_{m_k}), \tag{7}$$

where $J_{m_k}$ is the $m_k \times m_k$ identity matrix. In particular,

$$\log p_\phi(z_k|x_k) = -\frac{m_k}{2}\log 2\pi - \frac{1}{2}\log\sigma_k^2 - \frac{1}{2\sigma_k^2}\sum_{j=1}^{m_k}\{z_{k,j} - g_{\phi_k,j}(x_k)\}^2, \quad k = 1, 2. \tag{8}$$

The mean $g_{\phi_k}(x_k)$ can be any model, such as a neural network, with output of length $m_k$, $k = 1, 2$. The scale $\sigma_k$ can be fixed, computed via the weighted residuals, or learned through a separate set of models. It controls the amount of noise introduced in the latent dimension and is pivotal in determining the prediction interval width for $p(y|z)$. In this paper, $\sigma_k$ is fixed for simplicity.

**Decoder $p_\theta(y|z)$.** We assume a normal model for $Y$ given $Z = z$,

$$Y|Z = z \sim \mathcal{N}(f_\theta(z), \sigma^2). \tag{9}$$

This results in a log-likelihood contribution,

$$\log p_\theta(y|z) = -\frac{1}{2}\log 2\pi - \frac{1}{2}\log\sigma^2 - \frac{1}{2\sigma^2}\{y - f_\theta(z)\}^2. \tag{10}$$

Again, the mean $f_\theta(z)$ can be any model, such as a neural network.

**Model for missing data $p_\xi(x_1|x_2)$.** We use an empirical model for $X_1$ given $X_2 = x_2$ where $p_\xi(x_1|x_2)$ put masses only on those non-missing $x_1$'s in the training set. Let $I_{nm}$ be the set of indices such that $x_{1,j}$, $j \in I_{nm}$, are all the non-missing $x_1$ in the training set. Additionally, for a given $j \in I_{nm}$, $x_1[j]$ is the observed $x_1$ corresponding to $j$. For a given $x_2$, $p_\xi(x_2)$ is a vector of length $|I_{nm}|$, the cardinality of $I_{nm}$, with components

$$p_\xi(j|x_2) = \frac{\exp\{h_{\xi,j}(x_2)\}}{\sum_{t \in I_{nm}}\exp\{h_{\xi,t}(x_2)\}}, \quad j \in I_{nm}, \tag{11}$$

where $h_\xi$ is a vector of length $|I_{nm}|$, typically a neural network with input $x_2$ and output on $I_{nm}$. Now, the probability of $X_1 = x_1$ given $X_2 = x_2$ is

$$p_\xi(x_1|x_2) = \sum_{j \in I_{nm}} p_\xi(j|x_2) \cdot \mathbb{1}\{x_1 = x_1^{(nm)}[j]\}. \tag{12}$$

**Summary.** Overall, the M-step is

$$\phi_k^* = \arg\min_{\phi_k} \sum_{i,r} w_{i,r} \sum_{j=1}^{m_k} \{z_{k,i,r,j} - g_{\phi_k,j}(x_{k,i,r})\}^2, \quad k = 1, 2, \tag{13}$$

$$\theta^* = \arg\min_{\theta} \sum_{i,r} w_{i,r} \{y_i - f_\theta(z_{i,r})\}^2, \tag{14}$$

$$(\sigma^*)^2 = \frac{1}{N} \sum_{i,r} w_{i,r} \{y_i - f_{\theta^*}(z_{i,r})\}^2, \tag{15}$$

$$\xi^* = \arg\max_{\xi} \sum_{t \in D_x} w_t \log p_\xi(j_t | x_{2,t}). \tag{16}$$

When $x_{1,i}$ is missing, the sampling is enriched by simulated $x_{1,i,r}$. Eq. 16 selects $j^*$ from $I_{nm}$ based on $p_{\xi'}(j|x_2)$ according to Eq. 12 and Eq. 11. The $D_x$ above is a subset of the data for $\xi$, learned on the missing data part. We also note that learning the parameters above is parallelizable.

## 2.3 TRAINING

For efficiency purpose, the training set, $\{1, \ldots, N\}$, is segmented into batches $\{b_1, \ldots, b_L\}$ on which the index $i$ runs (and thus the denominator of Eq. 15 must be adapted accordingly). The process iterates for each batch until the maximum number of steps is reached or an early stopping criterion is satisfied. The full details are given in Algorithm 1 (Appendix A). The framework requires tuning hyper-parameters such as the number of MC samples $R$, the number of latent nodes $m_k$, and the standard deviation $\sigma_k$ of $Z_k$. Monitoring the point prediction on a hold-out validation is important to combat overfitting and terminate the training early with a PATIENCE hyper-parameter. Moreover, due to the generative nature of SEMF, the variation resulting from the initial random seed is measured in Subsection 4.2. Additionally, the model-specific hyper-parameters ($p_\phi$, $p_\theta$ and $p_\xi$) are also discussed in the same Subsection.

## 2.4 INFERENCE

The encoder-decoder structure of SEMF entails the simulations of $z_r$ during inference, as depicted in Figure 1. In theory, any inference can be performed for $\hat{y}$, for instance the mean value $\hat{y} = \frac{1}{R} \sum_{r=1}^{R} f_\theta(z_r)$, where $z_r \sim p_\phi(z|x)$ (see Algorithm 2 in Appendix A). For prediction intervals, a double simulation scheme is used,

$$z_r \sim p(z|x), \quad \hat{y}_{r,s} \sim p_\theta(y|z_r), \quad r, s = 1, \ldots, R. \tag{17}$$

Prediction interval at a given level of certainty $\alpha$ follows as

$$PI = \text{quantile}\left(\{\hat{y}_{r,s}\}; \frac{\alpha}{2}, 1 - \frac{\alpha}{2}\right). \tag{18}$$

**Remark.** We denote the $R$ for inference as $R_{\text{infer}}$.

# 3 RELATED WORK

## 3.1 LATENT REPRESENTATION LEARNING

Latent representation learning involves modeling hidden variables from observed data for various ML tasks, most notably Auto-Encoders (AEs) and VAEs. AEs are neural networks that reconstruct inputs by learning an intermediate latent representation. The encoder $g_\phi(\cdot)$ in an AE embeds the input $x$ into a latent variable $z$, which then passes through the decoder $f_\theta(\cdot)$, reconstructing $x$ as $\hat{x} = f_\theta(z)$. The training process optimizes the parameters $\phi$ and $\theta$ by minimizing the reconstruction loss. Unlike AEs, which focus on reconstruction, VAEs use variational methods to fit distributions of latent variables and the output (Kingma & Welling, 2014). Given a sample $x$ with a latent variable $z$, VAEs model the marginal likelihood $p_\theta(x)$. However, directly maximizing this likelihood is difficult (David M. Blei

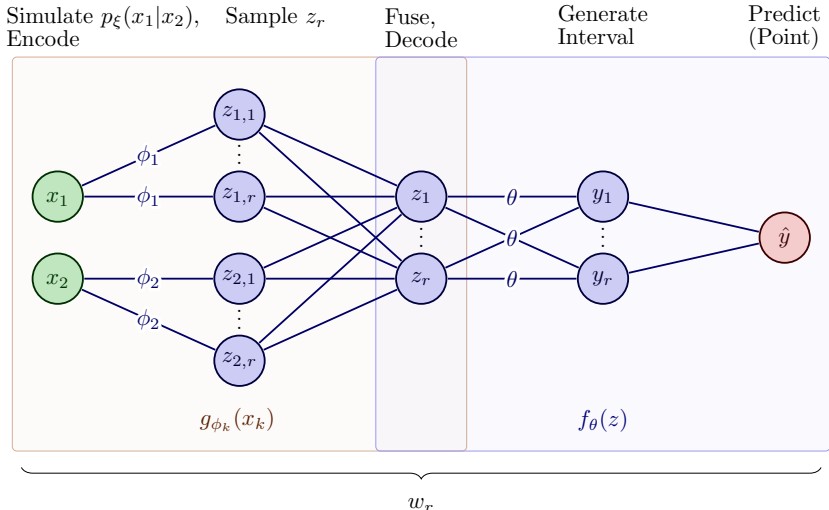

Figure 1: Inference procedure with the SEMF's learnable parameters $\phi_k$, $\theta$ and $\xi$. Here, we illustrate the number of inputs $k$ as $k = 1, 2$, assuming that $x_1$ may contain missing values

& McAuliffe, 2017b). Thus, alternative methods exist, such as maximizing the evidence lower bound (ELBO), which provides guarantees on the log-likelihood (Balakrishnan et al., 2017).

For supervised and semi-supervised tasks, latent representation learning can include task-specific predictions (Kingma et al., 2014). More specifically, models such as AEs follow the classical encoder-decoder objective while training a predictor $h_\psi(z)$ through an additional layer or model to estimate the output $y$. This dual objective helps in learning more task-relevant embeddings (Zhuang et al., 2015; Le et al., 2018). Semi-supervised VAEs are similar, with the distinction that they couple the reconstruction loss of the unlabeled data with a variational approximation of latent variables. This is effective even with sparse labels (Ji et al., 2020; Zhuang et al., 2023).

The EM algorithm has already been used for supervised learning tasks using specific models (Ghahramani & Jordan, 1993; Williams et al., 2005; Louiset et al., 2021), where the goal has been point prediction with GMMs. Similarly, the EM algorithm adapts well to minimal supervision (Luo et al., 2020) and using labeled and unlabeled data in semi-supervised settings for both single and multiple modalities (He & Jiang, 2022; Xu et al., 2024). Our work differs by modifying using MC sampling to generate prediction intervals with any ML model and, in theory, under any distribution.

## 3.2 PREDICTION INTERVALS

Crucial for estimating uncertainty, prediction intervals in regression are often derived using methods such as Bayesian approaches (Williams & Rasmussen, 1995; Hensman et al., 2015; Gal & Ghahramani, 2016), ensemble techniques (Breiman, 2001; Lakshminarayanan et al., 2017; Malinin et al., 2021), or quantile regression (Koenker & Bassett, 1978; Koenker & Hallock, 2001). Additionally, these methods can be complemented with conformal prediction, a framework for calibrating any point predictor to produce prediction intervals (Vovk et al., 2005; 2022), making it highly relevant for enhancing reliability in applications requiring rigorous uncertainty quantification.

A key component of quantile regression is the pinball loss function, which effectively balances the residuals to capture the desired quantiles even for non-parametric models (Steinwart & Christmann, 2011), making it ideal for asymmetric distributions where tail behavior is of critical importance (Koenker & Hallock, 2001). This loss function is pivotal not only for single model scenarios but also enhances ensemble methods by refining their quantile estimate (Meinshausen & Ridgeway, 2006). Conformal prediction further extends the applicability of these intervals by providing a layer of calibration that adjusts intervals obtained from any predictive model, ensuring they cover the true value with a pre-specified probability (Romano et al., 2019).

### 3.3 Missing Data

Managing missing values is a pivotal aspect when dealing with real-world data. Naive methods such as discarding instances or mean/median-imputation may be infeasible or carry the risk of changing the data distribution (Yoon et al., 2020; Jadhav et al., 2019). The chosen technique for handling missing data should adhere to the dataset's characteristics and mechanisms behind the missing data (Ibrahim et al., 2008). More advanced approaches, like the Iterative Imputer from scikit-learn (Pedregosa et al., 2011), an implementation of Multiple Imputation by Chained Equations (MICE) (van Buuren & Groothuis-Oorshoorn, 2011), expand on the simple imputations by iteratively modeling each feature with missing values as a function of other features (Buck, 1960; Schafer, 1997). Due to its complexity, MICE is best compared with alternatives such as K-means clustering and artificial neural networks (ANNs). K-means assigns missing values based on cluster centroids (Wang et al., 2019) as opposed to ANNs, which learn complex, non-linear relationships between variables (Pereira et al., 2020). ANNs effectively predict or reconstruct missing values and thus are particularly useful in datasets where relationships between variables are intricate and not easily captured by straightforward imputation methods. Accordingly, ANNs have demonstrated superior performance over MICE and GMM (with EM for missing values) in scenarios where a large proportion of the data is missing (Śmieja et al., 2018).

## 4 Experimental Setup

### 4.1 Datasets

We systematically curate a subset of datasets from the OpenML-CTR23 (Fischer et al., 2023) benchmark suite to evaluate and carry out our experiments. Initially comprising 35 datasets, we apply an exclusion criteria to refine this collection to 11 datasets. The details and overview are in Appendix B. We remove duplicated rows from all the datasets and carry out the standardization (scaling) of all predictors, including the outcome, which we transformed to have zero means and unit variances. The features of these datasets are then treated as separate inputs to SEMF. In the second stage of our experiments, we artificially introduce 50% missing values in our datasets for any predictor except for the first feature of a randomly chosen row, which emulates missing completely at random (MCAR) data. In all our datasets, 70% of the data is used to train all models, 15% as a hold-out validation set to monitor SEMF's performance, and 15% to evaluate the models. To combat overfitting, baseline models that benefit from early stopping are allocated another 15% from the training data. Lastly, it is essential to note that all data in SEMF are processed batch-wise, without employing mini-batch training, to ensure consistency and stability in the training process.

### 4.2 Models

Our baseline consists of both point and quantile regression eXtreme Gradient Boosting (XGBoost) (Chen & Guestrin, 2016a), Extremely Randomized Trees (ET) (Geurts et al., 2006), and neural networks (Tagasovska & Lopez-Paz, 2019), all summarized and depicted in Table 1. To ensure consistency in our experimental setup, we align the families and hyper-parameters of $p_\phi$ and $p_\theta$ with our baseline models. For example, in the case of XGBoost in SEMF, we use $K$ XGBoosts, $g_{\phi_k}(x_k)$, one for each input $x_k$, $k = 1, \ldots, K$, and one XGBoost for $f_\theta(z)$ with the same hyper-parameters. We refer to the SEMF's adoption of these models as MultiXGBs, MultiETs, and MultiMLPs. When establishing prediction intervals, we conformalize our prediction intervals according to Romano et al. (2019) at an uncertainty tolerance of 5% for both the baseline and SEMF (Eq. 18). The missing data simulator, $p_\xi$, is constructed using a shallow neural network, which employs the SELU activation function (Klambauer et al., 2017). It is experimented with two distinct node counts: 50 and 100.

To constrain the breadth of our parameter exploration, the simulator for the missing model adopts the optimal set of hyper-parameters identified from analyses involving complete datasets. We target (larger) $\sigma_k$ values that introduce more noise and produce better intervals than point predictions. The optimal models are then trained and tested with five different seeds, and the results are averaged. The point prediction, $\hat{y}$, uses the mean inferred values. We then study the performance of SEMF against mean and median imputation techniques, five nearest neighbors, and MICE from Pedregosa et al. (2011). The imputers are used within the point and interval baseline models explained in the

following subsection to form the missing value baseline. Appendix C contains more details on the hyper-parameters for each SEMF model and dataset.

Table 1: SEMF models, baselines, and hyper-parameters.

| SEMF | Point Prediction Baseline | Interval Prediction Baseline |
|---|---|---|
| MultiXGBs | XGBoost (Chen & Guestrin, 2016b) Trees: 100, Maximum depth: 6, Early stopping steps: 10 | Quantile XGBoost Same as point prediction baseline, XGBoost |
| MultiETs | Extremely Randomized Trees (Pedregosa et al., 2011) Trees: 100, Maximum depth: 10 | Quantile Extremely Randomized Trees (Johnson, 2024) Same as point prediction baseline, Extremely Randomized Trees |
| MultiMLPs | Deep Neural Network Hidden layers: 2, Nodes per layer: 100, Activation functions: ReLU, Epochs: 1000 or 5000, Learning rate: 0.001, Batch training, Early stopping steps: 100 | Simultaneous Quantile Regression (Tagasovska & Lopez-Paz, 2019) Same as point prediction baseline, Deep Neural Network |

### 4.3 METRICS

The evaluation of point predictions employs Root Mean Squared Error (RMSE), Mean Absolute Error (MAE), and R-squared ($R^2$). For prediction intervals, the chosen metrics, following Pearce et al. (2018) and Zhou et al. (2023), are the Prediction Interval Coverage Probability (PICP), and Normalized Mean Prediction Interval Width (NMPIW) as described in Appendix D.1. This paper also evaluates SEMF on our new metric, termed Coverage-Width Ratio (CWR),

$$\text{CWR} = \frac{\text{PICP}}{\text{NMPIW}}, \tag{19}$$

which evaluates the coverage probability ratio to the prediction interval's width. CWR provides a refined understanding of the balance between an interval's accuracy (coverage) and precision (width). Though a larger value of this metric is better in higher confidence levels, the marginal increase in NMPIW is likely higher than that of PICP, resulting in decreasing CWR.

In our case, measuring the performance of SEMF over the baseline models is far more critical than reviewing absolute metrics in isolation. For any metric above, except for ($R^2$), this is computed as

$$\text{Metric}_\Delta(\%) = \left( \frac{\text{Metric}_{\text{SEMF}} - \text{Metric}_{\text{Baseline}}}{\text{Metric}_{\text{Baseline}}} \right) \times 100, \tag{20}$$

on which we base our decisions for selecting the best hyper-parameters as explained in Appendix D.2.

## 5 RESULTS

We trained and tested 330 models corresponding to the three model types—MultiXGBs, MultiETs, and MultiMLPs—across 11 datasets with both complete and 50% missing data, using five seeds for each combination. Table 2 and Table 3 present the mean and standard deviation for our metrics aggregated over the five seeds. Appendix E includes the results from each individual run. For comparison, we also present the non-conformalized prediction intervals in Appendix F, though we solely discuss the results for conformalized intervals on both complete and incomplete data.

### 5.1 COMPLETE DATA

The results of models with complete data are in Table 2. Overall, MultiXGBs and MultiMLPs performed generally well in producing intervals compared to their baselines, as shown by positive ΔCWR and ΔNMPIW while achieving similar ΔPICP to the baselines. Notably, all models attained

Table 2: Test results for all models with complete data at 95% quantiles aggregated over five seeds. For each metric, the mean and standard deviation of the performance across the seeds are separated by ±. Performance over the baseline is highlighted in bold.

| | INTERVAL PREDICTIONS | | | | | POINT PREDICTIONS | | |
| | RELATIVE | | | ABSOLUTE | | RELATIVE | | ABSOLUTE |
| DATASET | ΔCWR | ΔPICP | ΔNMPIW | PICP | NMPIW | ΔRMSE | ΔMAE | $R^2$ |
|---|---|---|---|---|---|---|---|---|
| **MULTIXGBs** | | | | | | | | |
| SPACE_GA | **6%±3%** | -1%±0% | **7%±2%** | 0.94±0.01 | 0.26±0.01 | -9%±1% | -10%±2% | 0.60±0.01 |
| CPU_ACTIVITY | **16%±6%** | **1%±1%** | **12%±5%** | 0.94±0.01 | 0.09±0.00 | **21%±1%** | -1%±2% | 0.98±0.00 |
| NAVAL_PROPULSION_PLANT | **172%±14%** | 0%±1% | **63%±2%** | 0.95±0.01 | 0.11±0.00 | -14%±4% | -12%±2% | 0.99±0.00 |
| MIAMI_HOUSING | -7%±3% | 0%±0% | -8%±3% | 0.95±0.00 | 0.13±0.00 | **4%±4%** | **3%±3%** | 0.91±0.01 |
| KIN8NM | **6%±2%** | **1%±0%** | **5%±2%** | 0.94±0.01 | 0.45±0.01 | -20%±1% | -22%±1% | 0.63±0.01 |
| CONCRETE_COMPRESSIVE_STRENGTH | **41%±12%** | -3%±2% | **31%±7%** | 0.94±0.02 | 0.31±0.01 | -26%±6% | -40%±11% | 0.85±0.01 |
| CARS | **40%±18%** | -3%±1% | **30%±8%** | 0.91±0.01 | 0.13±0.01 | -3%±3% | -1%±3% | 0.95±0.00 |
| ENERGY_EFFICIENCY | **222%±45%** | -4%±3% | **70%±3%** | 0.92±0.02 | 0.05±0.01 | -15%±22% | -16%±17% | 1.00±0.00 |
| CALIFORNIA_HOUSING | -1%±3% | 0%±0% | -2%±4% | 0.95±0.00 | 0.42±0.01 | -1%±1% | -1%±1% | 0.81±0.00 |
| AIRFOIL_SELF_NOISE | **21%±21%** | -1%±2% | **15%±15%** | 0.97±0.02 | 0.36±0.06 | -64%±32% | -73%±41% | 0.86±0.05 |
| QSAR_FISH_TOXICITY | **34%±8%** | -5%±3% | **29%±5%** | 0.87±0.02 | 0.33±0.01 | **3%±3%** | **1%±4%** | 0.55±0.02 |
| **MULTIETs** | | | | | | | | |
| SPACE_GA | -6%±3% | **1%±1%** | -7%±3% | 0.96±0.00 | 0.29±0.01 | -15%±2% | -18%±3% | 0.54±0.02 |
| CPU_ACTIVITY | **11%±2%** | -4%±0% | **13%±2%** | 0.94±0.00 | 0.11±0.00 | -14%±1% | -20%±2% | 0.98±0.00 |
| NAVAL_PROPULSION_PLANT | **137%±22%** | **1%±0%** | **57%±4%** | 0.96±0.00 | 0.27±0.02 | -316%±45% | -406%±43% | 0.96±0.01 |
| MIAMI_HOUSING | -9%±1% | -1%±0% | -9%±1% | 0.95±0.00 | 0.15±0.00 | -10%±2% | -20%±2% | 0.90±0.00 |
| KIN8NM | -10%±1% | **1%±1%** | -12%±2% | 0.94±0.01 | 0.53±0.01 | -34%±2% | -38%±2% | 0.48±0.01 |
| CONCRETE_COMPRESSIVE_STRENGTH | -8%±5% | -3%±2% | -6%±6% | 0.89±0.02 | 0.31±0.02 | -67%±6% | -94%±8% | 0.76±0.01 |
| CARS | -25%±5% | 0%±3% | -34%±14% | 0.92±0.02 | 0.15±0.01 | **3%±4%** | **1%±2%** | 0.95±0.00 |
| ENERGY_EFFICIENCY | **10%±6%** | -4%±2% | **12%±6%** | 0.94±0.02 | 0.06±0.00 | **3%±2%** | **1%±1%** | 1.00±0.00 |
| CALIFORNIA_HOUSING | **1%±2%** | -2%±0% | **3%±2%** | 0.95±0.00 | 0.55±0.01 | -15%±1% | -21%±1% | 0.71±0.01 |
| AIRFOIL_SELF_NOISE | -16%±10% | -3%±2% | -17%±12% | 0.96±0.02 | 0.43±0.03 | -118%±43% | -141%±49% | 0.80±0.08 |
| QSAR_FISH_TOXICITY | -6%±9% | -5%±2% | -2%±12% | 0.88±0.02 | 0.36±0.02 | -6%±1% | -11%±1% | 0.53±0.01 |
| **MULTIMLPs** | | | | | | | | |
| SPACE_GA | **6%±3%** | -1%±1% | **6%±2%** | 0.95±0.01 | 0.23±0.00 | 0%±1% | -1%±1% | 0.75±0.01 |
| CPU_ACTIVITY | -7%±6% | 0%±1% | -9%±7% | 0.95±0.00 | 0.10±0.00 | **7%±2%** | **5%±2%** | 0.98±0.00 |
| NAVAL_PROPULSION_PLANT | **4%±16%** | **1%±0%** | 0%±14% | 0.96±0.00 | 0.08±0.01 | -45%±25% | -36%±20% | 1.00±0.00 |
| MIAMI_HOUSING | -38%±3% | 0%±1% | -61%±9% | 0.95±0.00 | 0.15±0.01 | -3%±2% | **4%±2%** | 0.91±0.00 |
| KIN8NM | **8%±5%** | 0%±1% | **8%±5%** | 0.95±0.01 | 0.20±0.01 | **7%±2%** | **7%±2%** | 0.93±0.00 |
| CONCRETE_COMPRESSIVE_STRENGTH | **11%±6%** | -2%±2% | **11%±6%** | 0.94±0.02 | 0.29±0.03 | **12%±5%** | **15%±4%** | 0.91±0.01 |
| CARS | **3%±12%** | -3%±3% | **4%±11%** | 0.92±0.02 | 0.13±0.01 | -3%±3% | 0%±3% | 0.95±0.00 |
| ENERGY_EFFICIENCY | **53%±21%** | 0%±4% | **34%±10%** | 0.96±0.02 | 0.05±0.00 | **32%±3%** | **33%±3%** | 1.00±0.00 |
| CALIFORNIA_HOUSING | -12%±3% | 0%±1% | -14%±4% | 0.95±0.00 | 0.43±0.01 | **7%±0%** | **9%±1%** | 0.82±0.00 |
| AIRFOIL_SELF_NOISE | **68%±28%** | -2%±1% | **40%±11%** | 0.97±0.01 | 0.18±0.01 | **18%±4%** | **16%±5%** | 0.97±0.00 |
| QSAR_FISH_TOXICITY | **5%±10%** | **1%±1%** | **3%±10%** | 0.89±0.03 | 0.35±0.04 | **4%±5%** | **7%±4%** | 0.55±0.04 |

suitable intervals on *naval_propulsion_plant* and *energy_efficiency*. Interestingly, MultiMLPs also attained good relative performance improvements for point prediction, which can indicate that the chosen $\sigma_k$ was too low for generating performant prediction intervals. MultiETs attained mixed results, with significant improvements in ΔCWR for other datasets such as *naval_propulsion_plant*, but its overall performance remains less conclusive.

## 5.2 MISSING DATA

The results of models with 50% missing data are in Table 3. Note that relative metrics for SEMF are compared with the best result from any baseline imputer on that metric, regardless of how the imputer performed on the other metrics. Overall, the results are worse and less consistent than with complete data. MultiXGBs maintained good performance on some datasets, such as *naval_propulsion_plant* and *energy_efficiency*, but declined on others, like *cpu_activity*. MultiETs continued to exhibit mixed results, with only *naval_propulsion_plant* offering marginally better performance over the baseline. Similarly, MultiMLPs worked well on only one dataset, namely *concrete_compressive_strength*, while the other datasets had increased model uncertainty.

## 5.3 DISCUSSION

Our results indicate that SEMF, when combined with XGBoost and MLPs, performs strongly on datasets with complete data. Both models produce better prediction intervals than traditional quantile regression methods in the complete case. MultiMLPs also deliver good point predictions for some datasets despite the experimental design prioritizing interval estimation over point accuracy. This could indicate either the effectiveness of the chosen $\sigma_k$, which (indirectly) led to narrower prediction

Table 3: Test results for all models with missing data at 95% quantiles aggregated over five seeds. For each metric, the mean and standard deviation of the performance across the seeds are separated by ±. Performance over the baseline is highlighted in bold.

| | INTERVAL PREDICTIONS | | | | | POINT PREDICTIONS | | |
| | RELATIVE | | | ABSOLUTE | | RELATIVE | | ABSOLUTE |
| DATASET | ΔCWR | ΔPICP | ΔNMPIW | PICP | NMPIW | ΔRMSE | ΔMAE | $R^2$ |
|---|---|---|---|---|---|---|---|---|
| **MultiXGBs** | | | | | | | | |
| SPACE_GA | 0%±3% | -2%±1% | **2%±3%** | 0.94±0.01 | 0.32±0.01 | -7%±4% | -9%±3% | 0.45±0.06 |
| CPU_ACTIVITY | -14%±9% | 0%±2% | -20%±14% | 0.94±0.03 | 0.18±0.02 | -1%±21% | -17%±13% | 0.83±0.07 |
| NAVAL_PROPULSION_PLANT | **18%±23%** | -3%±2% | **12%±18%** | 0.93±0.04 | 0.59±0.17 | -14%±20% | -29%±35% | 0.77±0.05 |
| MIAMI_HOUSING | -18%±13% | -1%±1% | -25%±22% | 0.94±0.01 | 0.20±0.02 | -14%±11% | -18%±10% | 0.72±0.10 |
| KIN8NM | -9%±3% | **1%±1%** | -13%±5% | 0.96±0.01 | 0.63±0.03 | -3%±3% | -4%±4% | 0.40±0.02 |
| CONCRETE_COMPRESSIVE_STRENGTH | **1%±9%** | -2%±2% | **1%±10%** | 0.95±0.01 | 0.57±0.02 | -16%±12% | -20%±14% | 0.58±0.08 |
| CARS | **9%±27%** | -7%±4% | **6%±23%** | 0.90±0.04 | 0.34±0.02 | -30%±22% | -35%±24% | 0.66±0.15 |
| ENERGY_EFFICIENCY | **26%±24%** | -4%±4% | **18%±18%** | 0.95±0.04 | 0.27±0.05 | -186%±265% | -153%±188% | 0.91±0.08 |
| CALIFORNIA_HOUSING | -6%±6% | -1%±1% | -8%±8% | 0.95±0.02 | 0.59±0.03 | -4%±4% | -5%±5% | 0.69±0.05 |
| AIRFOIL_SELF_NOISE | -24%±6% | -1%±2% | -32%±12% | 0.95±0.03 | 0.76±0.07 | -37%±31% | -43%±36% | 0.41±0.12 |
| QSAR_FISH_TOXICITY | 0%±8% | -4%±5% | 0%±11% | 0.91±0.05 | 0.50±0.11 | -4%±4% | -2%±3% | 0.36±0.04 |
| **MultiETs** | | | | | | | | |
| SPACE_GA | -11%±2% | 0%±2% | -14%±5% | 0.95±0.01 | 0.34±0.02 | -12%±4% | -13%±5% | 0.40±0.06 |
| CPU_ACTIVITY | -15%±8% | -3%±2% | -18%±13% | 0.94±0.02 | 0.20±0.02 | -21%±18% | -44%±14% | 0.82±0.07 |
| NAVAL_PROPULSION_PLANT | **4%±16%** | -4%±3% | **4%±18%** | 0.95±0.03 | 0.73±0.13 | -37%±21% | -95%±48% | 0.71±0.06 |
| MIAMI_HOUSING | -33%±10% | 0%±3% | -55%±24% | 0.95±0.01 | 0.27±0.02 | -25%±10% | -39%±14% | 0.68±0.12 |
| KIN8NM | -14%±1% | **1%±1%** | -17%±2% | 0.95±0.01 | 0.66±0.02 | -11%±2% | -13%±3% | 0.32±0.03 |
| CONCRETE_COMPRESSIVE_STRENGTH | -20%±3% | -2%±4% | -27%±9% | 0.95±0.02 | 0.61±0.02 | -37%±18% | -51%±22% | 0.48±0.06 |
| CARS | -32%±12% | -4%±4% | -53%±32% | 0.93±0.02 | 0.38±0.05 | -38%±15% | -32%±14% | 0.62±0.16 |
| ENERGY_EFFICIENCY | -30%±28% | -4%±2% | -82%±120% | 0.96±0.03 | 0.32±0.22 | -69%±60% | -75%±43% | 0.94±0.03 |
| CALIFORNIA_HOUSING | -6%±4% | -2%±1% | -4%±4% | 0.96±0.01 | 0.70±0.03 | -14%±3% | -17%±3% | 0.61±0.05 |
| AIRFOIL_SELF_NOISE | -31%±4% | -3%±1% | -43%±9% | 0.94±0.02 | 0.75±0.05 | -62%±51% | -89%±66% | 0.35±0.10 |
| QSAR_FISH_TOXICITY | -7%±7% | -3%±3% | -7%±6% | 0.92±0.04 | 0.50±0.08 | -9%±9% | -10%±9% | 0.39±0.09 |
| **MultiMLPs** | | | | | | | | |
| SPACE_GA | -25%±11% | -3%±2% | -34%±25% | 0.94±0.01 | 0.38±0.06 | -40%±23% | -34%±15% | 0.21±0.23 |
| CPU_ACTIVITY | -40%±9% | 0%±2% | -73%±24% | 0.95±0.01 | 0.25±0.02 | -15%±19% | -23%±17% | 0.75±0.12 |
| NAVAL_PROPULSION_PLANT | -43%±11% | -1%±2% | -89%±44% | 0.95±0.02 | 1.33±0.19 | -108%±66% | -183%±195% | -0.03±0.47 |
| MIAMI_HOUSING | -36%±7% | -1%±1% | -57%±20% | 0.95±0.02 | 0.22±0.02 | -25%±8% | -28%±11% | 0.67±0.10 |
| KIN8NM | -19%±3% | 0%±0% | -24%±5% | 0.96±0.01 | 0.70±0.02 | -16%±8% | -15%±9% | 0.37±0.04 |
| CONCRETE_COMPRESSIVE_STRENGTH | **5%±10%** | -3%±2% | **5%±10%** | 0.94±0.02 | 0.54±0.04 | -12%±5% | -12%±6% | 0.54±0.05 |
| CARS | -22%±17% | -1%±4% | -38%±30% | 0.95±0.03 | 0.35±0.06 | -28%±12% | -26%±13% | 0.66±0.10 |
| ENERGY_EFFICIENCY | -7%±11% | -4%±3% | -11%±14% | 0.95±0.03 | 0.31±0.08 | **1%±9%** | -1%±16% | 0.95±0.03 |
| CALIFORNIA_HOUSING | -24%±5% | 0%±1% | -34%±10% | 0.96±0.01 | 0.68±0.02 | -5%±2% | -3%±3% | 0.61±0.06 |
| AIRFOIL_SELF_NOISE | -8%±6% | -4%±3% | -7%±8% | 0.93±0.04 | 0.61±0.07 | -15%±8% | -9%±8% | 0.55±0.18 |
| QSAR_FISH_TOXICITY | -11%±5% | -2%±3% | -14%±8% | 0.92±0.04 | 0.53±0.11 | -13%±7% | -14%±8% | 0.34±0.06 |

intervals or underfitting of the MLPs. The truth may lie somewhere in between; SEMF's sampling operation, akin to cross-validation, partially helps combat overfitting, ensuring robust results despite variations in the data. In our preliminary experiments, we observed that increasing the depth of the baseline MLPs did not help and eventually led to overfitting. ETs exhibit more mixed performance, highly dependent on the dataset's characteristics. One possible explanation behind the larger variations in predictive power compared to other models may be ETs' reliance on randomized splits, which introduces significant variability in the prediction process. Consequently, ETs may not benefit from the iterative sampling and refinement of predictions inherent in SEMF.

The robustness of SEMF diminishes when applied to datasets with missing data. Despite not offering improvements over the baseline in the presence of missing data, we have presented these results to transparently illustrate our framework's current capabilities of handling missing inputs. From a theoretical standpoint, we believe there is value in further investigating how a single loss function that leverages EM's missing data capabilities can be effectively applied. Given that SEMF performs well with complete data, a practical alternative might involve using the best imputation method for the data at hand before applying SEMF, effectively treating the data as complete. We expect our ablation study to perform well, given the complete data results.

An important observation is the stability of SEMF across different random seeds, as indicated by consistent PICP and NMPIW metrics (full results in Appendix E), contrasting the significant variability observed in the baseline models. This suggests that SEMF offers a more reliable performance framework. Additionally, it is worth noting that our experiments did not precisely tune for conformalized prediction intervals but used the non-conformalized (raw) quantiles from SEMF. The non-conformalized intervals offer less coverage but produce better CWR and NMPIW than the baseline (Appendix F). Certain experimental choices—such as processing the columns

separately, using the same hyperparameters for both complete and incomplete data, and fixing the number of latent nodes per input ($m_k$)—may have further constrained the models' ability to tailor their performance to the varying complexities of different datasets.

## 6    CONCLUSION

This paper introduces the Supervised Expectation-Maximization Framework (SEMF), a novel model-agnostic approach for generating prediction intervals in datasets with missing values. SEMF draws from the EM algorithm for supervised learning to devise latent representations that produce better prediction intervals than quantile regression. Due to SEMF's iterative simulation technique, training and inference can be done with complete and incomplete data. A comprehensive set of 330 experimental runs on 11 datasets with three different model types showed that SEMF, in the case of complete data, outperforms quantile regression, particularly on complete datasets and when using XGBoost, which intrinsically lacks latent representations. The results of the missing data are less positive and require further investigation. This research underscores SEMF's potential in various application domains and opens new avenues for further exploration of supervised latent representation learning and uncertainty estimation.

## 7    LIMITATIONS & FUTURE WORK

The primary limitation of this study was its reliance on the normality assumption, which may not fully capture the potential of SEMF across diverse data distributions. Although in Appendix G we demonstrate that the framework can learn non-normal patterns, further investigation and exploration of SEMF under other distributions, such as uniform, log-normal, and generalized extreme value distributions, are needed. The computational complexity of the approach presents another significant challenge, as the current implementation can be optimized for large-scale applications. Additionally, while the CWR metric is useful, it implicitly assumes that a 1% drop in PICP equates to a 1% reduction in NMPIW, thus assuming a uniform distribution. Evaluating CWR under various distributional assumptions would provide a more comprehensive assessment of its implications. Additionally, SEMF has only been evaluated on MCAR data and does not address missing at random cases (MAR), which require further investigation for real-world applicability. Finally, applying the same hyper-parameters across all datasets without specific tuning for incomplete data likely contributes to the observed decline in accuracy and robustness.

Future work presents several intriguing avenues for exploration. A promising direction is the application of SEMF in multi-modal data settings, where the distinct $p_\phi$ components of the framework could be adapted to process diverse data types—from images and text to tabular datasets—enabling a more nuanced and powerful approach to integrating heterogeneous data sources. This capability positions the framework as a versatile tool for addressing missing data challenges across various domains and can also help expand it to discrete and multiple outputs. Another valuable area for development is the exploration of methods to capture and leverage dependencies among input features, which could improve the model's predictive performance and provide deeper insights into the underlying data structure. These advancements can enhance the broader appeal of end-to-end approaches like SEMF in the ML community.

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

# A  SEMF ALGORITHM

---

**Algorithm 1** SEMF Training: two input sources where $x_1$ can be missing

---

**Require:** $y, x_1, x_2, R$
**Ensure:** $\theta, \phi_1, \phi_2, \xi$
 1: Initialize $\theta, \phi_1, \phi_2, \xi$
 2: Initialize $D_y, D_{z_1}, D_{z_2}, D_x, D_i$ to $\emptyset$
 3: Split $I = \{1, \ldots, N\}$ into $L$ batches $\{b_1, \ldots, b_L\}$
 4: **for** $\ell = 1, \ldots, L$ **do**
 5:    **for all** $i$ in $b_\ell$ **do**
 6:       **if** $x_{1,i}$ is absent **then**
 7:          **for** $r = 1, \ldots, R$ **do**
 8:             Simulate $[j_{i,r}, x_{1,i,r}] \sim p_\xi(\cdot|x_{2,i})$
 9:             Simulate $z_{1,i,r} \sim p_{\phi_1}(\cdot|x_{1,i,r})$
10:             Simulate $z_{2,i,r} \sim p_{\phi_2}(\cdot|x_{2,i})$
11:             Set $z_{i,r} = [z_{1,i,r}, z_{2,i,r}]$
12:          **end for**
13:       **else**
14:          **for** $r = 1, \ldots, R$ **do**
15:             Simulate $z_{1,i,r} \sim p_{\phi_1}(\cdot|x_{1,i})$
16:             Simulate $z_{2,i,r} \sim p_{\phi_2}(\cdot|x_{2,i})$
17:             Set $z_{i,r} = [z_{1,i,r}, z_{2,i,r}]$
18:          **end for**
19:       **end if**
20:       **for** $r = 1, \ldots, R$ **do**
21:          Compute
$$w_{i,r} = \frac{p_\theta(y_i|z_{i,r})}{\sum_{t=1}^R p_\theta(y_i|z_{i,t})}$$
22:          Update $D_y \leftarrow D_y \cup [y_i|z_{i,r}|w_{i,r}]$
23:          Update $D_{z_2} \leftarrow D_{z_2} \cup [z_{2,i,r}|x_{2,i}|w_{i,r}]$
24:          **if** $x_{1,i}$ is absent **then**
25:             Update $D_{z_1} \leftarrow D_{z_1} \cup [z_{1,i,r}|x_{1,i,r}|w_{i,r}]$
26:             Update $D_x \leftarrow D_x \cup [j_{i,r}|x_{2,i}|w_{i,r}]$
27:          **else**
28:             Update $D_{z_1} \leftarrow D_{z_1} \cup [z_{1,i,r}|x_{1,i}|w_{i,r}]$
29:          **end if**
30:       **end for**
31:       Update $\theta \leftarrow Q_y(\theta, D_y)$
32:       Update $\phi_1 \leftarrow Q_1(\phi_1, D_{z_1})$
33:       Update $\phi_2 \leftarrow Q_2(\phi_2, D_{z_2})$
34:       Update $\xi \leftarrow Q_x(\xi, D_x)$
35:    **end for**
36: **end for**
37: Check convergence; Go to step 4 if not

---

---

**Algorithm 2** SEMF Inference

---

**Require:** $\theta^*, \phi_1^*, \phi_2^*, \xi^*, x_1, x_2, R$
**Ensure:** $z_{i,r}$
  1: **for** $r = 1, \ldots, R$ **do**
  2:   **if** $x_{1,i}$ is absent **then**
  3:     Simulate $[j_{i,r}, x_{1,i,r}] \sim p_{\xi^*}(\cdot|x_{2,i})$
  4:     Simulate $z_{1,i,r} \sim p_{\phi_1^*}(\cdot|x_{1,i,r})$
  5:     Simulate $z_{2,i,r} \sim p_{\phi_2^*}(\cdot|x_{2,i})$
  6:     Set $z_{i,r} = [z_{1,i,r}, z_{2,i,r}]$
  7:   **else**
  8:     Simulate $z_{1,i,r} \sim p_{\phi_1^*}(\cdot|x_{1,i})$
  9:     Simulate $z_{2,i,r} \sim p_{\phi_2^*}(\cdot|x_{2,i})$
 10:     Set $z_{i,r} = [z_{1,i,r}, z_{2,i,r}]$
 11:   **end if**
 12: **end for**

---

## B  DATASETS FOR TABULAR BENCHMARK

OpenML-CTR23 (Fischer et al., 2023) datasets are selected in the following manner. The first criterion is to exclude datasets exceeding 30,000 instances or 30 features to maintain computational tractability. Moreover, we exclude the *moneyball* data (Kaggle, 2017) to control for missing values and any datasets with non-numeric features, such as those with temporal or ordinal data not encoded numerically. We then categorize the datasets based on size: small for those with less than ten features, medium for 10 to 19 features, and large for 20 to 29 features. We apply a similar size classification based on the number of instances, considering datasets with more than 10,000 instances as large. To avoid computational constraints, we exclude datasets that were large in both features and instances, ensuring a varied yet manageable set for our experiments. This leads us to the final list of 11 datasets listed in Table 4.

Table 4: Summary of benchmark tabular datasets retained from (Fischer et al., 2023)

| DATASET NAME | N SAMPLES | N FEATURES | OPENML DATA ID | Y [MIN:MAX] | SOURCE |
|---|---|---|---|---|---|
| SPACE_GA | 3,107 | 7 | 45402 | [-3.06:0.1] | (PACE & BARRY, 1997) |
| CPU_ACTIVITY | 8,192 | 22 | 44978 | [0:99] | (RASMUSSEN ET AL., 1996) |
| NAVAL_PROPULSION_PLANT | 11,934 | 15 | 44969 | [0.95:1.0] | (CORADDU ET AL., 2016) |
| MIAMI_HOUSING | 13,932 | 16 | 44983 | [72,000:2,650,000] | (KAGGLE, 2022) |
| KIN8NM | 8,192 | 9 | 44980 | [0.04:1.46] | (GHAHRAMANI, 1996) |
| CONCRETE_COMPRESSIVE_STRENGTH | 1,030 | 9 | 44959 | [2.33:82.6] | (YEH, 1998) |
| CARS | 804 | 18 | 44994 | [8,639:70,756] | (KUIPER, 2008) |
| ENERGY_EFFICIENCY | 768 | 9 | 44960 | [6.01:43.1] | (TSANAS & XIFARA, 2012) |
| CALIFORNIA_HOUSING | 20,640 | 9 | 44977 | [14,999:500,001] | (KELLEY PACE & BARRY, 1997) |
| AIRFOIL_SELF_NOISE | 1,503 | 6 | 44957 | [103.38:140.98] | (BROOKS ET AL., 1989) |
| QSAR_FISH_TOXICITY | 908 | 7 | 44970 | [0.053:9.612] | (M. CASSOTTI & CONSONNI, 2015) |

## C  OPTIMAL SET OF HYPER-PARAMETERS

The hyper-parameter tuning for SEMF is implemented and monitored using Weights & Biases (Biewald, 2020). A random search is done in the hyper-parameter space for a maximum of 500 iterations on all 11 datasets, focusing on tuning the models only on the complete datasets. Key hyper-parameters are varied across a predefined set to balance accuracy and computational efficiency. The following grid is used for hyper-parameter tuning: the number of importance sampling operations $R \in \{5, 10, 25, 50, 100\}$ (100 is omitted for MultiMLPS), nodes per latent dimension $m_k \in \{1, 5, 10, 20, 30\}$, and standard deviations $\sigma_{m_k} \in \{0.001, 0.01, 0.1, 1.0\}$. Early stopping steps (PATIENCE) are set to five or ten, and $R_{\text{infer}}$ is explored at [30, 50, 70]. The option to run the models in parallel must be consistently enabled. Table 5 shows the optimal set of hyper-parameters. This table includes common hyper-parameters for complete and 50% datasets and another part showing $\xi_{\text{nodes}}$, which is tuned manually and only relevant to the missing data.

Table 5: Hyper-parameters for MultiXGBs, MultiETs, and MultiMLPs used for both complete and missing data.

| Dataset | Complete and Missing | | | | | Missing |
|---|---|---|---|---|---|---|
| | R | $m_k$ | $\sigma_k$ | Patience | $R_{\text{infer}}$ | $\xi_{\text{nodes}}$ |
| **MultiXGBs** | | | | | | |
| SPACE_GA | 10 | 30 | 1.0 | 5 | 70 | 100 |
| CPU_ACTIVITY | 5 | 30 | 1.0 | 5 | 70 | 50 |
| NAVAL_PROPULSION_PLANT | 5 | 30 | 0.01 | 5 | 50 | 100 |
| MIAMI_HOUSING | 5 | 10 | 0.1 | 5 | 50 | 50 |
| KIN8NM | 5 | 30 | 1.0 | 10 | 70 | 100 |
| CONCRETE_COMPRESSIVE_STRENGTH | 25 | 30 | 1.0 | 5 | 70 | 100 |
| CARS | 50 | 10 | 1.0 | 10 | 70 | 100 |
| ENERGY_EFFICIENCY | 5 | 1 | 0.01 | 10 | 70 | 100 |
| CALIFORNIA_HOUSING | 5 | 10 | 0.1 | 10 | 50 | 100 |
| AIRFOIL_SELF_NOISE | 25 | 1 | 0.01 | 10 | 70 | 100 |
| QSAR_FISH_TOXICITY | 50 | 30 | 1.0 | 5 | 70 | 100 |
| **MultiETs** | | | | | | |
| SPACE_GA | 10 | 30 | 1.0 | 10 | 70 | 100 |
| CPU_ACTIVITY | 5 | 30 | 1.0 | 10 | 70 | 50 |
| NAVAL_PROPULSION_PLANT | 5 | 30 | 0.01 | 10 | 50 | 100 |
| MIAMI_HOUSING | 10 | 10 | 0.1 | 10 | 50 | 50 |
| KIN8NM | 5 | 30 | 1.0 | 10 | 70 | 100 |
| CONCRETE_COMPRESSIVE_STRENGTH | 25 | 30 | 1.0 | 10 | 70 | 100 |
| CARS | 100 | 5 | 0.1 | 10 | 100 | 100 |
| ENERGY_EFFICIENCY | 5 | 1 | 0.01 | 10 | 70 | 100 |
| CALIFORNIA_HOUSING | 5 | 10 | 0.1 | 5 | 50 | 100 |
| AIRFOIL_SELF_NOISE | 25 | 1 | 0.01 | 10 | 70 | 100 |
| QSAR_FISH_TOXICITY | 50 | 30 | 1.0 | 10 | 70 | 100 |
| **MultiMLPs** | | | | | | |
| SPACE_GA | 25 | 10 | 0.001 | 10 | 50 | 100 |
| CPU_ACTIVITY | 5 | 20 | 0.001 | 5 | 50 | 50 |
| NAVAL_PROPULSION_PLANT | 5 | 20 | 0.001 | 5 | 50 | 100 |
| MIAMI_HOUSING | 5 | 20 | 0.01 | 5 | 50 | 50 |
| KIN8NM | 5 | 20 | 0.001 | 5 | 50 | 100 |
| CONCRETE_COMPRESSIVE_STRENGTH | 5 | 30 | 0.001 | 10 | 50 | 100 |
| CARS | 5 | 30 | 0.1 | 5 | 50 | 100 |
| ENERGY_EFFICIENCY | 50 | 30 | 0.1 | 10 | 50 | 100 |
| CALIFORNIA_HOUSING | 5 | 20 | 0.01 | 5 | 50 | 100 |
| AIRFOIL_SELF_NOISE | 25 | 10 | 0.01 | 10 | 50 | 100 |
| QSAR_FISH_TOXICITY | 50 | 30 | 1.0 | 10 | 70 | 100 |

MultiXGBs and MultiMLPs benefit from early stopping to reduce computation time in complete and incomplete cases. Similarly, the baseline models for these instances use the same hyper-parameters for early stopping. Further, the number of epochs in the case of MultiMLPs is set as 1000, except for *energy_efficiency* and *QSAR_fish_toxicity*, where this is changed to 5000. Any model-specific hyperparameter we did not specify in this paper remains at the implementation's default value (e.g., the number of leaves in XGBoost from Chen & Guestrin (2016b)). Along with the supplementary code, we provide three additional CSV files: one for the results and hyperparameters of all 330 runs and the other two for the optimal hyperparameters of SEMF models, both raw (directly from SEMF) and conformalized.

Additional conditions are applied only for experiments with missing data. Datasets *california_housing*, *cpu_activity*, *miami_housing*, and *naval_propulsion_plant*—have a PATIENCE of five to expedite the training process. Additionally, for *california_housing* and *cpu_activity*, the $R_{\text{infer}}$ value is set to 30,

while for all the other datasets, it is set to 50. We do this to ensure efficient computation, speed, and memory usage (especially for the GPU).

For training MultiXGBs and MultiETs, the computations are performed in parallel using CPU cores (Intel® Core™ i9-13900KF). For MultiMLPs, they are done on a GPU (NVIDIA® GeForce RTX™ 4090). The GPU is also consistently used for the missing data simulator and training $p_\xi$. All the computations are done on a machine with 32 GB of memory. The code provides further details on hardware and reproducibility.

## D   METRICS FOR PREDICTION INTERVALS

### D.1   COMMON METRICS

The most common metrics for evaluating prediction intervals (Pearce et al., 2018; Zhou et al., 2023) are:

- Prediction Interval Coverage Probability (PICP): This metric assesses the proportion of times the true value of the target variable falls within the constructed prediction intervals. For a set of test examples $(x_1, y_1), \ldots, (x_N, y_N)$, a given level of confidence $\alpha$, and their corresponding prediction intervals $I_1, \ldots, I_N$, the PICP is calculated as:

$$\text{PICP} = \frac{1}{N} \sum_{i=1}^{N} \mathbb{1}(y_i \in [L_i, U_i]), \tag{21}$$

where $U_i$ and $L_i$ are the upper and lower bounds of the predicted values for the $i$-th instance. $y_i$ is the actual value of the $i$-th test example, and $\mathbb{1}$ is the indicator function, which equals 1 if $y_i$ is in the interval $[L_i, U_i]$ and 0 otherwise. $0 \le \text{PICP} \le 1$ where PICP closer to 1 and higher than the confidence level $\alpha$ is favored.

- Mean Prediction Interval Width (MPIW): The average width is computed as

$$\text{MPIW} = \frac{1}{N} \sum_{i=1}^{N} (U_i - L_i), \tag{22}$$

which shows the sharpness or uncertainty, where $0 \le \text{MPIW} < \infty$ and MPIW close to 0 is preferred.

- Normalized Mean Prediction Interval Width (NMPIW): Since MPIW varies by dataset, it can be normalized by the range of the target variable

$$\text{NMPIW} = \frac{\text{MPIW}}{\max(y) - \min(y)} \tag{23}$$

where $\max(y)$ and $\min(y)$ are the maximum and minimum values of the target variable, respectively. The interpretation remains the same as MPIW.

## D.2 IMPACT OF RELATIVE METRICS FOR MODELING

As our primary focus is on interval prediction, configurations demonstrating the most significant improvements in ΔCWR and ΔPICP are prioritized when selecting the optimal hyper-parameters. Furthermore, both ΔPICP and ΔCWR must be positive, indicating that we must at least have the same reliability of the baseline (PICP) with better or same interval ratios (CWR). In instances where no configuration meets the initial improvement criteria for both metrics, we relax the requirement for positive ΔPICP to accept values greater than -5% and subsequently -10%, allowing us to consider configurations where SEMF significantly improves CWR, even if the PICP improvement is less marked but remains within an acceptable range for drawing comparisons.

## E   Full conformalized results

Table 6: Test results for MultiXGBs with complete data at 95% quantiles for seeds 0, 10, 20, 30, 40, with rows ordered by seed (ascending). Performance over the baseline is highlighted in bold.

| | Interval Predictions | | | | | | Point Predictions | | |
| | Relative | | | Absolute | | | Relative | | Absolute |
| Dataset | ΔCWR | ΔPICP | ΔNMPIW | PICP | MPIW | NMPIW | ΔRMSE | ΔMAE | R$^2$ |
|---|---|---|---|---|---|---|---|---|---|
| SPACE_GA | **2%** | -2% | **4%** | 0.95 | 2.40 | 0.26 | -10% | -13% | 0.60 |
| SPACE_GA | **10%** | -1% | **10%** | 0.95 | 2.34 | 0.25 | -10% | -10% | 0.62 |
| SPACE_GA | **8%** | -2% | **9%** | 0.93 | 2.24 | 0.24 | -9% | -8% | 0.60 |
| SPACE_GA | **6%** | -2% | **7%** | 0.94 | 2.38 | 0.26 | -8% | -10% | 0.59 |
| SPACE_GA | **7%** | -1% | **7%** | 0.95 | 2.55 | 0.28 | -8% | -10% | 0.59 |
| CPU_ACTIVITY | **7%** | **1%** | **5%** | 0.95 | 0.49 | 0.09 | **22%** | **2%** | 0.98 |
| CPU_ACTIVITY | **20%** | **1%** | **17%** | 0.93 | 0.45 | 0.09 | **20%** | -5% | 0.98 |
| CPU_ACTIVITY | **20%** | 0% | **17%** | 0.93 | 0.45 | 0.09 | **21%** | 0% | 0.98 |
| CPU_ACTIVITY | **21%** | **1%** | **16%** | 0.94 | 0.45 | 0.09 | **23%** | -2% | 0.98 |
| CPU_ACTIVITY | **10%** | **2%** | **7%** | 0.95 | 0.47 | 0.09 | **20%** | -3% | 0.98 |
| NAVAL_PROPULSION_PLANT | **163%** | -1% | **62%** | 0.95 | 0.36 | 0.11 | -15% | -12% | 0.99 |
| NAVAL_PROPULSION_PLANT | **190%** | 0% | **66%** | 0.95 | 0.37 | 0.11 | -20% | -14% | 0.99 |
| NAVAL_PROPULSION_PLANT | **151%** | 0% | **60%** | 0.96 | 0.40 | 0.12 | -12% | -9% | 0.99 |
| NAVAL_PROPULSION_PLANT | **185%** | -1% | **65%** | 0.94 | 0.35 | 0.10 | -9% | -15% | 0.99 |
| NAVAL_PROPULSION_PLANT | **169%** | -1% | **63%** | 0.96 | 0.37 | 0.11 | -15% | -11% | 0.99 |
| MIAMI_HOUSING | -2% | 0% | -2% | 0.95 | 0.99 | 0.12 | **4%** | **1%** | 0.91 |
| MIAMI_HOUSING | -10% | 0% | -10% | 0.95 | 1.03 | 0.13 | **7%** | **7%** | 0.90 |
| MIAMI_HOUSING | -6% | **1%** | -8% | 0.95 | 0.99 | 0.12 | **9%** | **5%** | 0.91 |
| MIAMI_HOUSING | -7% | 0% | -8% | 0.95 | 0.99 | 0.12 | **3%** | -1% | 0.91 |
| MIAMI_HOUSING | -10% | 0% | -12% | 0.95 | 1.05 | 0.13 | -4% | 0% | 0.90 |
| KIN8NM | **6%** | 0% | **6%** | 0.94 | 2.27 | 0.45 | -21% | -24% | 0.62 |
| KIN8NM | **4%** | **1%** | **3%** | 0.95 | 2.39 | 0.47 | -19% | -21% | 0.64 |
| KIN8NM | **9%** | **1%** | **8%** | 0.94 | 2.20 | 0.43 | -17% | -20% | 0.64 |
| KIN8NM | **9%** | **1%** | **7%** | 0.95 | 2.36 | 0.46 | -21% | -22% | 0.63 |
| KIN8NM | **3%** | **1%** | **2%** | 0.94 | 2.33 | 0.46 | -20% | -23% | 0.62 |
| CONCRETE_COMPRESSIVE_STRENGTH | **40%** | -3% | **31%** | 0.93 | 1.42 | 0.29 | -16% | -22% | 0.86 |
| CONCRETE_COMPRESSIVE_STRENGTH | **47%** | -6% | **36%** | 0.91 | 1.48 | 0.30 | -29% | -52% | 0.84 |
| CONCRETE_COMPRESSIVE_STRENGTH | **59%** | -2% | **39%** | 0.96 | 1.60 | 0.32 | -21% | -34% | 0.86 |
| CONCRETE_COMPRESSIVE_STRENGTH | **35%** | -3% | **28%** | 0.95 | 1.48 | 0.30 | -28% | -39% | 0.83 |
| CONCRETE_COMPRESSIVE_STRENGTH | **24%** | 0% | **19%** | 0.97 | 1.59 | 0.32 | -34% | -52% | 0.86 |
| CARS | **56%** | -4% | **38%** | 0.91 | 0.70 | 0.12 | -7% | -4% | 0.95 |
| CARS | **36%** | -5% | **30%** | 0.90 | 0.86 | 0.14 | -4% | -4% | 0.95 |
| CARS | **30%** | -3% | **25%** | 0.91 | 0.76 | 0.13 | **2%** | **5%** | 0.96 |
| CARS | **65%** | -1% | **40%** | 0.93 | 0.75 | 0.13 | -5% | -1% | 0.95 |
| CARS | **16%** | -4% | **17%** | 0.91 | 0.87 | 0.15 | -2% | 0% | 0.95 |
| ENERGY_EFFICIENCY | **165%** | -5% | **64%** | 0.92 | 0.20 | 0.06 | -50% | -29% | 1.00 |
| ENERGY_EFFICIENCY | **288%** | 0% | **74%** | 0.91 | 0.14 | 0.04 | -3% | -13% | 1.00 |
| ENERGY_EFFICIENCY | **217%** | -1% | **69%** | 0.96 | 0.18 | 0.05 | -21% | -31% | 1.00 |
| ENERGY_EFFICIENCY | **253%** | -3% | **73%** | 0.92 | 0.16 | 0.04 | **17%** | **16%** | 1.00 |
| ENERGY_EFFICIENCY | **185%** | -10% | **68%** | 0.91 | 0.23 | 0.07 | -17% | -22% | 1.00 |
| CALIFORNIA_HOUSING | -5% | 0% | -6% | 0.95 | 1.82 | 0.43 | -2% | -3% | 0.81 |
| CALIFORNIA_HOUSING | **4%** | 0% | **4%** | 0.95 | 1.72 | 0.41 | -1% | -2% | 0.81 |
| CALIFORNIA_HOUSING | -4% | **1%** | -5% | 0.95 | 1.76 | 0.42 | 0% | -1% | 0.82 |
| CALIFORNIA_HOUSING | **1%** | 0% | **1%** | 0.95 | 1.74 | 0.41 | 0% | 0% | 0.81 |
| CALIFORNIA_HOUSING | -3% | **1%** | -4% | 0.95 | 1.79 | 0.42 | -1% | -1% | 0.82 |
| AIRFOIL_SELF_NOISE | **15%** | -4% | **16%** | 0.95 | 1.73 | 0.37 | -82% | -93% | 0.86 |
| AIRFOIL_SELF_NOISE | **45%** | **1%** | **30%** | 0.97 | 1.42 | 0.30 | -18% | -18% | 0.93 |
| AIRFOIL_SELF_NOISE | **5%** | **1%** | **4%** | 0.98 | 1.86 | 0.40 | -73% | -76% | 0.85 |
| AIRFOIL_SELF_NOISE | -6% | 0% | -7% | 0.98 | 2.15 | 0.46 | -109% | -138% | 0.78 |
| AIRFOIL_SELF_NOISE | **46%** | -2% | **33%** | 0.95 | 1.37 | 0.29 | -41% | -43% | 0.89 |
| QSAR_FISH_TOXICITY | **21%** | -2% | **19%** | 0.88 | 2.12 | 0.33 | **3%** | **3%** | 0.53 |
| QSAR_FISH_TOXICITY | **39%** | -9% | **35%** | 0.85 | 1.97 | 0.31 | **8%** | **7%** | 0.58 |
| QSAR_FISH_TOXICITY | **43%** | -3% | **32%** | 0.89 | 2.21 | 0.34 | **4%** | **3%** | 0.57 |
| QSAR_FISH_TOXICITY | **38%** | -3% | **30%** | 0.89 | 2.11 | 0.33 | -2% | -1% | 0.54 |
| QSAR_FISH_TOXICITY | **29%** | -6% | **28%** | 0.85 | 2.11 | 0.33 | 0% | -6% | 0.52 |

Table 7: Test results for MultiETs with complete data at 95% quantiles for seeds 0, 10, 20, 30, 40, with rows ordered by seed (ascending). Performance over the baseline is highlighted in bold.

| | INTERVAL PREDICTIONS | | | | | | POINT PREDICTIONS | | |
| | RELATIVE | | | ABSOLUTE | | | RELATIVE | | ABSOLUTE |
| DATASET | ΔCWR | ΔPICP | ΔNMPIW | PICP | MPIW | NMPIW | ΔRMSE | ΔMAE | $R^2$ |
|---|---|---|---|---|---|---|---|---|---|
| SPACE_GA | -8% | **2%** | -11% | 0.95 | 2.59 | 0.28 | -13% | -15% | 0.55 |
| SPACE_GA | -3% | 0% | -3% | 0.95 | 2.77 | 0.30 | -14% | -17% | 0.55 |
| SPACE_GA | -2% | **1%** | -4% | 0.96 | 2.61 | 0.28 | -13% | -13% | 0.57 |
| SPACE_GA | -9% | **1%** | -11% | 0.96 | 2.81 | 0.30 | -15% | -18% | 0.54 |
| SPACE_GA | -7% | 0% | -8% | 0.96 | 2.79 | 0.30 | -19% | -24% | 0.51 |
| CPU_ACTIVITY | **11%** | -4% | **13%** | 0.95 | 0.57 | 0.11 | -13% | -20% | 0.98 |
| CPU_ACTIVITY | **14%** | -5% | **16%** | 0.94 | 0.54 | 0.10 | -13% | -18% | 0.98 |
| CPU_ACTIVITY | **9%** | -4% | **11%** | 0.95 | 0.58 | 0.11 | -13% | -18% | 0.98 |
| CPU_ACTIVITY | **10%** | -4% | **13%** | 0.94 | 0.57 | 0.11 | -15% | -22% | 0.97 |
| CPU_ACTIVITY | **10%** | -4% | **13%** | 0.94 | 0.57 | 0.11 | -16% | -22% | 0.97 |
| NAVAL_PROPULSION_PLANT | **131%** | **1%** | **56%** | 0.96 | 0.96 | 0.28 | -320% | -418% | 0.96 |
| NAVAL_PROPULSION_PLANT | **144%** | **1%** | **59%** | 0.96 | 0.88 | 0.26 | -286% | -369% | 0.97 |
| NAVAL_PROPULSION_PLANT | **174%** | 0% | **63%** | 0.96 | 0.81 | 0.24 | -257% | -348% | 0.97 |
| NAVAL_PROPULSION_PLANT | **127%** | 0% | **56%** | 0.96 | 0.94 | 0.28 | -324% | -426% | 0.96 |
| NAVAL_PROPULSION_PLANT | **108%** | **1%** | **52%** | 0.96 | 1.03 | 0.30 | -391% | -469% | 0.95 |
| MIAMI_HOUSING | -7% | 0% | -7% | 0.95 | 1.22 | 0.15 | -11% | -20% | 0.90 |
| MIAMI_HOUSING | -10% | -1% | -10% | 0.95 | 1.24 | 0.15 | -6% | -17% | 0.90 |
| MIAMI_HOUSING | -8% | -1% | -8% | 0.95 | 1.25 | 0.15 | -13% | -23% | 0.89 |
| MIAMI_HOUSING | -9% | -1% | -10% | 0.95 | 1.22 | 0.15 | -9% | -19% | 0.90 |
| MIAMI_HOUSING | -9% | 0% | -9% | 0.94 | 1.24 | 0.15 | -10% | -22% | 0.89 |
| KIN8NM | -10% | **1%** | -12% | 0.95 | 2.75 | 0.54 | -36% | -40% | 0.46 |
| KIN8NM | -8% | 0% | -9% | 0.94 | 2.66 | 0.52 | -33% | -36% | 0.49 |
| KIN8NM | -11% | **3%** | -15% | 0.95 | 2.76 | 0.54 | -32% | -35% | 0.50 |
| KIN8NM | -9% | **2%** | -12% | 0.94 | 2.67 | 0.52 | -36% | -40% | 0.48 |
| KIN8NM | -10% | **1%** | -13% | 0.95 | 2.73 | 0.53 | -33% | -36% | 0.49 |
| CONCRETE_COMPRESSIVE_STRENGTH | -18% | -3% | -18% | 0.89 | 1.67 | 0.34 | -78% | -110% | 0.74 |
| CONCRETE_COMPRESSIVE_STRENGTH | -5% | -3% | -3% | 0.89 | 1.51 | 0.30 | -71% | -97% | 0.76 |
| CONCRETE_COMPRESSIVE_STRENGTH | -6% | -6% | 0% | 0.87 | 1.42 | 0.29 | -63% | -87% | 0.77 |
| CONCRETE_COMPRESSIVE_STRENGTH | -6% | -3% | -4% | 0.90 | 1.47 | 0.30 | -63% | -91% | 0.77 |
| CONCRETE_COMPRESSIVE_STRENGTH | -6% | **1%** | -7% | 0.92 | 1.59 | 0.32 | -61% | -88% | 0.77 |
| CARS | -35% | **5%** | -60% | 0.95 | 1.01 | 0.17 | **6%** | **3%** | 0.95 |
| CARS | -25% | **1%** | -34% | 0.91 | 0.83 | 0.14 | **8%** | **4%** | 0.95 |
| CARS | -19% | 0% | -24% | 0.92 | 0.80 | 0.14 | **4%** | -2% | 0.95 |
| CARS | -24% | -3% | -28% | 0.90 | 0.86 | 0.15 | **2%** | 0% | 0.95 |
| CARS | -21% | -3% | -22% | 0.91 | 0.91 | 0.15 | -3% | **1%** | 0.94 |
| ENERGY_EFFICIENCY | **16%** | -8% | **21%** | 0.90 | 0.18 | 0.05 | 0% | 0% | 1.00 |
| ENERGY_EFFICIENCY | **6%** | -2% | **6%** | 0.97 | 0.21 | 0.06 | **4%** | **1%** | 1.00 |
| ENERGY_EFFICIENCY | **8%** | -4% | **11%** | 0.95 | 0.21 | 0.06 | **5%** | **2%** | 1.00 |
| ENERGY_EFFICIENCY | **3%** | -3% | **5%** | 0.96 | 0.21 | 0.06 | **1%** | 0% | 1.00 |
| ENERGY_EFFICIENCY | **17%** | -4% | **18%** | 0.95 | 0.19 | 0.06 | **5%** | 0% | 1.00 |
| CALIFORNIA_HOUSING | -2% | -2% | **1%** | 0.95 | 2.38 | 0.57 | -16% | -22% | 0.70 |
| CALIFORNIA_HOUSING | **3%** | -2% | **5%** | 0.95 | 2.31 | 0.55 | -13% | -19% | 0.72 |
| CALIFORNIA_HOUSING | **2%** | -2% | **4%** | 0.95 | 2.29 | 0.55 | -15% | -20% | 0.71 |
| CALIFORNIA_HOUSING | 0% | -3% | **3%** | 0.95 | 2.32 | 0.55 | -16% | -22% | 0.71 |
| CALIFORNIA_HOUSING | **1%** | -3% | **4%** | 0.95 | 2.32 | 0.55 | -14% | -20% | 0.72 |
| AIRFOIL_SELF_NOISE | -25% | -5% | -26% | 0.94 | 2.17 | 0.46 | -166% | -184% | 0.74 |
| AIRFOIL_SELF_NOISE | 0% | -1% | **1%** | 0.98 | 1.78 | 0.38 | -59% | -80% | 0.89 |
| AIRFOIL_SELF_NOISE | -15% | -1% | -16% | 0.98 | 1.98 | 0.42 | -97% | -106% | 0.84 |
| AIRFOIL_SELF_NOISE | -28% | -5% | -32% | 0.93 | 2.20 | 0.47 | -169% | -213% | 0.67 |
| AIRFOIL_SELF_NOISE | -14% | -3% | -13% | 0.96 | 1.93 | 0.41 | -98% | -125% | 0.84 |
| QSAR_FISH_TOXICITY | -16% | -2% | -17% | 0.88 | 2.45 | 0.38 | -6% | -10% | 0.53 |
| QSAR_FISH_TOXICITY | -10% | -2% | -9% | 0.89 | 2.43 | 0.38 | -7% | -12% | 0.52 |
| QSAR_FISH_TOXICITY | **3%** | -6% | **9%** | 0.88 | 2.38 | 0.37 | -8% | -12% | 0.50 |
| QSAR_FISH_TOXICITY | -12% | -4% | -9% | 0.88 | 2.41 | 0.38 | -5% | -10% | 0.55 |
| QSAR_FISH_TOXICITY | **8%** | -9% | **16%** | 0.84 | 2.03 | 0.32 | -6% | -13% | 0.53 |

Table 8: Test results for MultiMLPs with complete data at 95% quantiles for seeds 0, 10, 20, 30, 40, with rows ordered by seed (ascending). Performance over the baseline is highlighted in bold.

| | INTERVAL PREDICTIONS | | | | | | POINT PREDICTIONS | | |
| | RELATIVE | | | ABSOLUTE | | | RELATIVE | | ABSOLUTE |
| DATASET | ΔCWR | ΔPICP | ΔNMPIW | PICP | MPIW | NMPIW | ΔRMSE | ΔMAE | R$^2$ |
|---|---|---|---|---|---|---|---|---|---|
| SPACE_GA | **5%** | -2% | **7%** | 0.95 | 2.11 | 0.23 | 0% | -1% | 0.74 |
| SPACE_GA | **3%** | -1% | **3%** | 0.96 | 2.13 | 0.23 | -1% | -1% | 0.75 |
| SPACE_GA | **6%** | -1% | **7%** | 0.94 | 2.04 | 0.22 | -3% | -3% | 0.74 |
| SPACE_GA | **5%** | -2% | **6%** | 0.95 | 2.15 | 0.23 | 0% | 0% | 0.76 |
| SPACE_GA | **10%** | 0% | **9%** | 0.95 | 2.06 | 0.22 | **1%** | 0% | 0.76 |
| CPU_ACTIVITY | -18% | **1%** | -22% | 0.96 | 0.58 | 0.11 | **4%** | **2%** | 0.98 |
| CPU_ACTIVITY | -3% | 0% | -2% | 0.95 | 0.51 | 0.10 | **8%** | **7%** | 0.98 |
| CPU_ACTIVITY | -9% | 0% | -10% | 0.95 | 0.53 | 0.10 | **6%** | **5%** | 0.98 |
| CPU_ACTIVITY | -6% | 0% | -7% | 0.95 | 0.54 | 0.10 | **6%** | **5%** | 0.98 |
| CPU_ACTIVITY | -2% | 0% | -2% | 0.95 | 0.53 | 0.10 | **8%** | **5%** | 0.98 |
| NAVAL_PROPULSION_PLANT | **32%** | **1%** | **24%** | 0.95 | 0.23 | 0.07 | -44% | -48% | 1.00 |
| NAVAL_PROPULSION_PLANT | 0% | **1%** | -2% | 0.96 | 0.31 | 0.09 | -9% | -11% | 1.00 |
| NAVAL_PROPULSION_PLANT | **7%** | 0% | **5%** | 0.95 | 0.24 | 0.07 | -35% | -13% | 1.00 |
| NAVAL_PROPULSION_PLANT | -8% | **1%** | -11% | 0.96 | 0.28 | 0.08 | -86% | -63% | 1.00 |
| NAVAL_PROPULSION_PLANT | -13% | **2%** | -16% | 0.96 | 0.34 | 0.10 | -51% | -45% | 0.99 |
| MIAMI_HOUSING | -38% | 0% | -60% | 0.95 | 1.18 | 0.15 | -6% | **1%** | 0.91 |
| MIAMI_HOUSING | -37% | 0% | -59% | 0.95 | 1.16 | 0.14 | -2% | **7%** | 0.91 |
| MIAMI_HOUSING | -43% | 0% | -74% | 0.95 | 1.25 | 0.15 | -4% | **6%** | 0.91 |
| MIAMI_HOUSING | -38% | **1%** | -62% | 0.95 | 1.15 | 0.14 | -2% | **3%** | 0.92 |
| MIAMI_HOUSING | -33% | -1% | -47% | 0.95 | 1.10 | 0.14 | -1% | **4%** | 0.91 |
| KIN8NM | **10%** | **1%** | **9%** | 0.95 | 1.03 | 0.20 | **4%** | **5%** | 0.93 |
| KIN8NM | **2%** | **1%** | **1%** | 0.96 | 1.07 | 0.21 | **10%** | **10%** | 0.94 |
| KIN8NM | **4%** | **1%** | **3%** | 0.95 | 1.06 | 0.21 | **10%** | **6%** | 0.94 |
| KIN8NM | **10%** | -3% | **12%** | 0.93 | 0.99 | 0.19 | **6%** | **5%** | 0.93 |
| KIN8NM | **15%** | -1% | **13%** | 0.94 | 1.03 | 0.20 | **7%** | **7%** | 0.93 |
| CONCRETE_COMPRESSIVE_STRENGTH | **7%** | -3% | **9%** | 0.91 | 1.24 | 0.25 | **18%** | **19%** | 0.92 |
| CONCRETE_COMPRESSIVE_STRENGTH | **18%** | **1%** | **14%** | 0.97 | 1.39 | 0.28 | **9%** | **12%** | 0.91 |
| CONCRETE_COMPRESSIVE_STRENGTH | **2%** | **1%** | **1%** | 0.97 | 1.63 | 0.33 | **9%** | **15%** | 0.91 |
| CONCRETE_COMPRESSIVE_STRENGTH | **7%** | -5% | **11%** | 0.94 | 1.62 | 0.33 | **4%** | **10%** | 0.90 |
| CONCRETE_COMPRESSIVE_STRENGTH | **19%** | -3% | **18%** | 0.93 | 1.26 | 0.26 | **18%** | **20%** | 0.91 |
| CARS | **5%** | -9% | **13%** | 0.88 | 0.66 | 0.11 | -5% | -2% | 0.95 |
| CARS | 0% | -3% | **2%** | 0.93 | 0.80 | 0.14 | -6% | -2% | 0.95 |
| CARS | -9% | 0% | -10% | 0.93 | 0.80 | 0.14 | **1%** | **1%** | 0.96 |
| CARS | -8% | -3% | -4% | 0.93 | 0.83 | 0.14 | -2% | -1% | 0.95 |
| CARS | **25%** | -1% | **21%** | 0.94 | 0.79 | 0.13 | -4% | **5%** | 0.95 |
| ENERGY_EFFICIENCY | **76%** | -3% | **46%** | 0.94 | 0.16 | 0.04 | **31%** | **31%** | 1.00 |
| ENERGY_EFFICIENCY | **46%** | **7%** | **25%** | 0.99 | 0.19 | 0.06 | **35%** | **32%** | 1.00 |
| ENERGY_EFFICIENCY | **34%** | -1% | **27%** | 0.95 | 0.17 | 0.05 | **28%** | **32%** | 1.00 |
| ENERGY_EFFICIENCY | **29%** | -3% | **25%** | 0.95 | 0.16 | 0.05 | **31%** | **32%** | 1.00 |
| ENERGY_EFFICIENCY | **79%** | -3% | **45%** | 0.96 | 0.17 | 0.05 | **37%** | **38%** | 1.00 |
| CALIFORNIA_HOUSING | -13% | **1%** | -16% | 0.95 | 1.88 | 0.45 | **8%** | **11%** | 0.82 |
| CALIFORNIA_HOUSING | -8% | 0% | -9% | 0.95 | 1.78 | 0.42 | **7%** | **8%** | 0.81 |
| CALIFORNIA_HOUSING | -16% | **1%** | -20% | 0.95 | 1.83 | 0.44 | **7%** | **9%** | 0.82 |
| CALIFORNIA_HOUSING | -11% | -1% | -12% | 0.94 | 1.82 | 0.43 | **7%** | **8%** | 0.81 |
| CALIFORNIA_HOUSING | -12% | 0% | -13% | 0.94 | 1.76 | 0.42 | **7%** | **9%** | 0.82 |
| AIRFOIL_SELF_NOISE | **98%** | -2% | **51%** | 0.97 | 0.82 | 0.18 | **21%** | **19%** | 0.97 |
| AIRFOIL_SELF_NOISE | **33%** | 0% | **24%** | 0.98 | 0.90 | 0.19 | **17%** | **12%** | 0.97 |
| AIRFOIL_SELF_NOISE | **36%** | -3% | **28%** | 0.95 | 0.79 | 0.17 | **11%** | **9%** | 0.97 |
| AIRFOIL_SELF_NOISE | **83%** | -4% | **47%** | 0.95 | 0.76 | 0.16 | **22%** | **22%** | 0.97 |
| AIRFOIL_SELF_NOISE | **93%** | -1% | **49%** | 0.98 | 0.88 | 0.19 | **18%** | **18%** | 0.97 |
| QSAR_FISH_TOXICITY | -11% | **2%** | -16% | 0.91 | 2.53 | 0.39 | **5%** | **11%** | 0.51 |
| QSAR_FISH_TOXICITY | **3%** | **2%** | **1%** | 0.91 | 2.38 | 0.37 | 0% | -1% | 0.57 |
| QSAR_FISH_TOXICITY | **7%** | -1% | **7%** | 0.91 | 2.31 | 0.36 | **6%** | **11%** | 0.54 |
| QSAR_FISH_TOXICITY | **18%** | **2%** | **14%** | 0.90 | 2.10 | 0.33 | -4% | **6%** | 0.52 |
| QSAR_FISH_TOXICITY | **9%** | -1% | **9%** | 0.83 | 1.84 | 0.29 | **10%** | **9%** | 0.61 |

Table 9: Test results for MultiXGBs with 50% missing data at 95% quantiles for seeds 0, 10, 20, 30, 40, with rows ordered by seed (ascending). Performance over the baseline is highlighted in bold.

| | INTERVAL PREDICTIONS | | | | | | POINT PREDICTIONS | | |
| | RELATIVE | | | ABSOLUTE | | | RELATIVE | | ABSOLUTE |
| DATASET | ΔCWR | ΔPICP | ΔNMPIW | PICP | MPIW | NMPIW | ΔRMSE | ΔMAE | $R^2$ |
|---|---|---|---|---|---|---|---|---|---|
| SPACE_GA | **4%** | -4% | **7%** | 0.93 | 2.82 | 0.31 | -11% | -13% | 0.35 |
| SPACE_GA | **2%** | -1% | **3%** | 0.94 | 2.97 | 0.32 | -6% | -6% | 0.45 |
| SPACE_GA | -4% | -3% | -2% | 0.94 | 3.06 | 0.33 | -12% | -11% | 0.44 |
| SPACE_GA | -1% | -1% | -1% | 0.95 | 2.97 | 0.32 | -1% | -6% | 0.46 |
| SPACE_GA | **2%** | 0% | **2%** | 0.95 | 2.77 | 0.30 | -5% | -11% | 0.54 |
| CPU_ACTIVITY | -9% | **1%** | -11% | 0.95 | 0.89 | 0.17 | -7% | -23% | 0.94 |
| CPU_ACTIVITY | -26% | **1%** | -44% | 0.93 | 1.02 | 0.19 | -1% | -26% | 0.81 |
| CPU_ACTIVITY | -15% | -2% | -14% | 0.89 | 0.80 | 0.15 | -30% | -30% | 0.73 |
| CPU_ACTIVITY | -1% | -1% | -3% | 0.95 | 0.86 | 0.16 | **36%** | **5%** | 0.88 |
| CPU_ACTIVITY | -19% | **2%** | -26% | 0.98 | 1.12 | 0.21 | -2% | -11% | 0.81 |
| NAVAL_PROPULSION_PLANT | -8% | 0% | -11% | 0.98 | 2.93 | 0.86 | -45% | -90% | 0.73 |
| NAVAL_PROPULSION_PLANT | **31%** | -2% | **24%** | 0.93 | 1.50 | 0.44 | **5%** | -6% | 0.84 |
| NAVAL_PROPULSION_PLANT | **18%** | -2% | **17%** | 0.96 | 2.06 | 0.61 | -30% | -42% | 0.83 |
| NAVAL_PROPULSION_PLANT | **53%** | -5% | **37%** | 0.87 | 1.27 | 0.37 | **6%** | **6%** | 0.76 |
| NAVAL_PROPULSION_PLANT | -5% | -3% | -6% | 0.93 | 2.22 | 0.65 | -7% | -11% | 0.70 |
| MIAMI_HOUSING | -10% | -2% | -11% | 0.94 | 1.45 | 0.18 | -7% | -12% | 0.80 |
| MIAMI_HOUSING | -38% | 0% | -63% | 0.93 | 1.86 | 0.23 | -35% | -35% | 0.54 |
| MIAMI_HOUSING | -26% | -1% | -33% | 0.95 | 1.74 | 0.22 | -10% | -19% | 0.76 |
| MIAMI_HOUSING | **1%** | -3% | **2%** | 0.95 | 1.35 | 0.17 | -14% | -20% | 0.79 |
| MIAMI_HOUSING | -17% | **1%** | -22% | 0.94 | 1.64 | 0.20 | -4% | -4% | 0.72 |
| KIN8NM | -6% | 0% | -8% | 0.97 | 3.36 | 0.66 | 0% | **1%** | 0.40 |
| KIN8NM | -15% | **1%** | -22% | 0.97 | 3.46 | 0.68 | -6% | -8% | 0.39 |
| KIN8NM | -8% | **3%** | -12% | 0.95 | 3.00 | 0.59 | **1%** | **2%** | 0.41 |
| KIN8NM | -9% | **2%** | -14% | 0.96 | 3.11 | 0.61 | -5% | -7% | 0.44 |
| KIN8NM | -8% | **1%** | -9% | 0.95 | 3.23 | 0.63 | -5% | -6% | 0.37 |
| CONCRETE_COMPRESSIVE_STRENGTH | -8% | -3% | -8% | 0.95 | 2.81 | 0.57 | -19% | -22% | 0.48 |
| CONCRETE_COMPRESSIVE_STRENGTH | **7%** | -5% | **11%** | 0.93 | 2.79 | 0.56 | -37% | -44% | 0.57 |
| CONCRETE_COMPRESSIVE_STRENGTH | **14%** | -3% | **13%** | 0.97 | 2.69 | 0.55 | -4% | -9% | 0.64 |
| CONCRETE_COMPRESSIVE_STRENGTH | **4%** | -1% | **3%** | 0.95 | 2.83 | 0.57 | -15% | -24% | 0.50 |
| CONCRETE_COMPRESSIVE_STRENGTH | -11% | 0% | -14% | 0.97 | 3.01 | 0.61 | -4% | -3% | 0.69 |
| CARS | **31%** | -3% | **24%** | 0.93 | 1.91 | 0.32 | -9% | -16% | 0.75 |
| CARS | **51%** | -7% | **38%** | 0.93 | 1.84 | 0.31 | -37% | -25% | 0.81 |
| CARS | -16% | -11% | -17% | 0.89 | 2.17 | 0.37 | -68% | -80% | 0.55 |
| CARS | -13% | -3% | -21% | 0.94 | 2.18 | 0.37 | -8% | -14% | 0.76 |
| CARS | -5% | -12% | **5%** | 0.83 | 2.00 | 0.34 | -28% | -39% | 0.40 |
| ENERGY_EFFICIENCY | -11% | 0% | -13% | 1.00 | 1.15 | 0.33 | -15% | -24% | 0.97 |
| ENERGY_EFFICIENCY | **20%** | -10% | **25%** | 0.89 | 0.84 | 0.24 | -705% | -520% | 0.76 |
| ENERGY_EFFICIENCY | **42%** | -1% | **23%** | 0.95 | 0.78 | 0.23 | -37% | -39% | 0.94 |
| ENERGY_EFFICIENCY | **61%** | -4% | **40%** | 0.94 | 0.70 | 0.20 | -16% | -44% | 0.96 |
| ENERGY_EFFICIENCY | **20%** | -3% | **16%** | 0.97 | 1.12 | 0.33 | -155% | -137% | 0.93 |
| CALIFORNIA_HOUSING | -14% | 0% | -17% | 0.96 | 2.67 | 0.64 | -3% | -3% | 0.65 |
| CALIFORNIA_HOUSING | 0% | -2% | 0% | 0.96 | 2.41 | 0.57 | -11% | -13% | 0.73 |
| CALIFORNIA_HOUSING | -7% | -1% | -8% | 0.96 | 2.48 | 0.59 | 0% | **1%** | 0.73 |
| CALIFORNIA_HOUSING | **2%** | -3% | **3%** | 0.92 | 2.28 | 0.54 | -4% | -7% | 0.61 |
| CALIFORNIA_HOUSING | -12% | -1% | -16% | 0.97 | 2.63 | 0.63 | -1% | -3% | 0.73 |
| AIRFOIL_SELF_NOISE | -30% | -2% | -43% | 0.97 | 3.95 | 0.84 | -94% | -110% | 0.50 |
| AIRFOIL_SELF_NOISE | -18% | -1% | -22% | 0.91 | 2.98 | 0.64 | -19% | -29% | 0.38 |
| AIRFOIL_SELF_NOISE | -23% | -4% | -25% | 0.91 | 3.49 | 0.75 | -43% | -48% | 0.19 |
| AIRFOIL_SELF_NOISE | -31% | **1%** | -49% | 0.97 | 3.61 | 0.77 | -14% | -8% | 0.52 |
| AIRFOIL_SELF_NOISE | -17% | -1% | -21% | 0.97 | 3.81 | 0.81 | -14% | -22% | 0.43 |
| QSAR_FISH_TOXICITY | -8% | **3%** | -14% | 0.98 | 4.22 | 0.66 | **4%** | **4%** | 0.43 |
| QSAR_FISH_TOXICITY | -6% | **1%** | -7% | 0.97 | 3.93 | 0.61 | -9% | -3% | 0.35 |
| QSAR_FISH_TOXICITY | **8%** | -5% | **10%** | 0.88 | 2.59 | 0.40 | -4% | -5% | 0.38 |
| QSAR_FISH_TOXICITY | -5% | -5% | -5% | 0.88 | 2.83 | 0.44 | -7% | -3% | 0.35 |
| QSAR_FISH_TOXICITY | **12%** | -11% | **15%** | 0.85 | 2.58 | 0.40 | -5% | -3% | 0.30 |

Table 10: Test results for MultiETs with 50% missing data at 95% quantiles for seeds 0, 10, 20, 30, 40, with rows ordered by seed (ascending). Performance over the baseline is highlighted in bold.

| | INTERVAL PREDICTIONS | | | | | | POINT PREDICTIONS | | |
| | RELATIVE | | | ABSOLUTE | | | RELATIVE | | ABSOLUTE |
| DATASET | ΔCWR | ΔPICP | ΔNMPIW | PICP | MPIW | NMPIW | ΔRMSE | ΔMAE | R² |
|---|---|---|---|---|---|---|---|---|---|
| SPACE_GA | -7% | -2% | -6% | 0.94 | 3.08 | 0.33 | -18% | -19% | 0.29 |
| SPACE_GA | -12% | **3%** | -18% | 0.97 | 3.41 | 0.37 | -7% | -7% | 0.40 |
| SPACE_GA | -13% | 0% | -16% | 0.95 | 3.05 | 0.33 | -13% | -11% | 0.42 |
| SPACE_GA | -13% | 0% | -18% | 0.96 | 3.21 | 0.35 | -8% | -8% | 0.41 |
| SPACE_GA | -10% | 0% | -11% | 0.96 | 2.91 | 0.32 | -15% | -18% | 0.46 |
| CPU_ACTIVITY | -18% | -3% | -23% | 0.95 | 1.03 | 0.19 | -40% | -56% | 0.92 |
| CPU_ACTIVITY | -12% | -6% | -11% | 0.91 | 1.04 | 0.20 | -15% | -44% | 0.81 |
| CPU_ACTIVITY | -5% | -5% | -2% | 0.92 | 1.00 | 0.19 | -43% | -59% | 0.75 |
| CPU_ACTIVITY | -13% | -2% | -14% | 0.96 | 0.97 | 0.18 | **2%** | -19% | 0.87 |
| CPU_ACTIVITY | -30% | -1% | -42% | 0.96 | 1.31 | 0.25 | -10% | -43% | 0.76 |
| NAVAL_PROPULSION_PLANT | -19% | 0% | -23% | 0.98 | 3.17 | 0.94 | -63% | -158% | 0.71 |
| NAVAL_PROPULSION_PLANT | **11%** | -5% | **13%** | 0.94 | 2.24 | 0.66 | -31% | -78% | 0.73 |
| NAVAL_PROPULSION_PLANT | **14%** | -2% | **14%** | 0.97 | 2.26 | 0.67 | -61% | -147% | 0.80 |
| NAVAL_PROPULSION_PLANT | **24%** | -9% | **25%** | 0.89 | 1.99 | 0.59 | -13% | -42% | 0.66 |
| NAVAL_PROPULSION_PLANT | -10% | -2% | -11% | 0.95 | 2.75 | 0.81 | -19% | -52% | 0.64 |
| MIAMI_HOUSING | -15% | -2% | -17% | 0.94 | 1.86 | 0.23 | -11% | -25% | 0.79 |
| MIAMI_HOUSING | -44% | **5%** | -87% | 0.96 | 2.39 | 0.30 | -43% | -64% | 0.47 |
| MIAMI_HOUSING | -41% | 0% | -69% | 0.96 | 2.30 | 0.29 | -27% | -42% | 0.72 |
| MIAMI_HOUSING | -35% | -2% | -52% | 0.96 | 2.18 | 0.27 | -21% | -33% | 0.79 |
| MIAMI_HOUSING | -31% | **1%** | -48% | 0.95 | 2.24 | 0.28 | -22% | -30% | 0.64 |
| KIN8NM | -15% | -1% | -19% | 0.96 | 3.55 | 0.69 | -8% | -8% | 0.29 |
| KIN8NM | -14% | **2%** | -19% | 0.96 | 3.43 | 0.67 | -10% | -14% | 0.32 |
| KIN8NM | -11% | 0% | -13% | 0.94 | 3.20 | 0.63 | -9% | -12% | 0.32 |
| KIN8NM | -14% | 0% | -18% | 0.95 | 3.30 | 0.65 | -14% | -18% | 0.37 |
| KIN8NM | -14% | **2%** | -19% | 0.96 | 3.42 | 0.67 | -13% | -16% | 0.30 |
| CONCRETE_COMPRESSIVE_STRENGTH | -20% | **4%** | -35% | 0.96 | 3.01 | 0.61 | -30% | -43% | 0.41 |
| CONCRETE_COMPRESSIVE_STRENGTH | -16% | -8% | -10% | 0.92 | 3.08 | 0.62 | -72% | -94% | 0.47 |
| CONCRETE_COMPRESSIVE_STRENGTH | -18% | -3% | -24% | 0.96 | 2.99 | 0.60 | -21% | -33% | 0.56 |
| CONCRETE_COMPRESSIVE_STRENGTH | -25% | -4% | -34% | 0.94 | 2.86 | 0.58 | -26% | -36% | 0.41 |
| CONCRETE_COMPRESSIVE_STRENGTH | -22% | **2%** | -31% | 0.96 | 3.05 | 0.62 | -35% | -50% | 0.53 |
| CARS | -24% | -6% | -27% | 0.92 | 1.99 | 0.34 | -22% | -23% | 0.73 |
| CARS | -49% | **4%** | -103% | 0.97 | 2.70 | 0.46 | -20% | -17% | 0.80 |
| CARS | -14% | -5% | -12% | 0.93 | 2.02 | 0.34 | -59% | -56% | 0.60 |
| CARS | -36% | -6% | -54% | 0.93 | 2.44 | 0.41 | -37% | -37% | 0.66 |
| CARS | -37% | -6% | -68% | 0.89 | 2.12 | 0.36 | -50% | -25% | 0.34 |
| ENERGY_EFFICIENCY | **2%** | -3% | **5%** | 0.97 | 0.62 | 0.18 | -18% | -32% | 0.98 |
| ENERGY_EFFICIENCY | -42% | -6% | -62% | 0.94 | 0.85 | 0.25 | -159% | -104% | 0.90 |
| ENERGY_EFFICIENCY | -4% | -6% | **2%** | 0.93 | 0.69 | 0.20 | -8% | -20% | 0.95 |
| ENERGY_EFFICIENCY | -29% | -4% | -41% | 0.94 | 0.79 | 0.23 | -39% | -82% | 0.95 |
| ENERGY_EFFICIENCY | -75% | 0% | -316% | 1.00 | 2.61 | 0.76 | -122% | -136% | 0.92 |
| CALIFORNIA_HOUSING | -2% | -2% | 0% | 0.96 | 2.88 | 0.69 | -11% | -16% | 0.60 |
| CALIFORNIA_HOUSING | -9% | -2% | -8% | 0.96 | 2.90 | 0.69 | -19% | -23% | 0.64 |
| CALIFORNIA_HOUSING | -5% | -2% | -4% | 0.97 | 3.02 | 0.72 | -11% | -13% | 0.64 |
| CALIFORNIA_HOUSING | -1% | -3% | **1%** | 0.93 | 2.77 | 0.66 | -15% | -18% | 0.50 |
| CALIFORNIA_HOUSING | -10% | -1% | -10% | 0.97 | 3.07 | 0.73 | -13% | -17% | 0.64 |
| AIRFOIL_SELF_NOISE | -33% | -3% | -46% | 0.97 | 3.62 | 0.77 | -163% | -218% | 0.34 |
| AIRFOIL_SELF_NOISE | -36% | -2% | -56% | 0.93 | 3.42 | 0.73 | -27% | -36% | 0.42 |
| AIRFOIL_SELF_NOISE | -31% | -5% | -40% | 0.92 | 3.42 | 0.73 | -50% | -83% | 0.17 |
| AIRFOIL_SELF_NOISE | -32% | -2% | -45% | 0.97 | 3.91 | 0.83 | -45% | -57% | 0.43 |
| AIRFOIL_SELF_NOISE | -22% | -2% | -30% | 0.94 | 3.29 | 0.70 | -27% | -51% | 0.36 |
| QSAR_FISH_TOXICITY | **2%** | **1%** | 0% | 0.98 | 4.12 | 0.64 | **4%** | **2%** | 0.51 |
| QSAR_FISH_TOXICITY | -10% | -4% | -7% | 0.93 | 3.43 | 0.53 | -12% | -12% | 0.40 |
| QSAR_FISH_TOXICITY | -7% | **1%** | -11% | 0.93 | 2.87 | 0.45 | -6% | -8% | 0.45 |
| QSAR_FISH_TOXICITY | -17% | -7% | -16% | 0.86 | 2.99 | 0.47 | -23% | -26% | 0.25 |
| QSAR_FISH_TOXICITY | 0% | -3% | -1% | 0.90 | 2.63 | 0.41 | -4% | -6% | 0.34 |

Table 11: Test results for MultiMLPs with 50% missing data at 95% quantiles for seeds 0, 10, 20, 30, 40, with rows ordered by seed (ascending). Performance over the baseline is highlighted in bold.

| | INTERVAL PREDICTIONS | | | | | | POINT PREDICTIONS | | |
| | RELATIVE | | | ABSOLUTE | | | RELATIVE | | ABSOLUTE |
| DATASET | ΔCWR | ΔPICP | ΔNMPIW | PICP | MPIW | NMPIW | ΔRMSE | ΔMAE | $R^2$ |
|---|---|---|---|---|---|---|---|---|---|
| SPACE_GA | -25% | -3% | -32% | 0.95 | 3.45 | 0.37 | -41% | -40% | 0.15 |
| SPACE_GA | -22% | -3% | -27% | 0.96 | 3.58 | 0.39 | -28% | -25% | 0.35 |
| SPACE_GA | -45% | -2% | -82% | 0.95 | 4.52 | 0.49 | -85% | -61% | -0.21 |
| SPACE_GA | -13% | -2% | -13% | 0.95 | 3.12 | 0.34 | -21% | -20% | 0.37 |
| SPACE_GA | -20% | -6% | -18% | 0.92 | 3.07 | 0.33 | -25% | -26% | 0.38 |
| CPU_ACTIVITY | -22% | -2% | -32% | 0.95 | 1.14 | 0.21 | **14%** | **1%** | 0.92 |
| CPU_ACTIVITY | -44% | **2%** | -80% | 0.95 | 1.40 | 0.27 | -30% | -34% | 0.72 |
| CPU_ACTIVITY | -49% | **1%** | -100% | 0.94 | 1.44 | 0.27 | -37% | -49% | 0.54 |
| CPU_ACTIVITY | -45% | **2%** | -89% | 0.96 | 1.23 | 0.23 | -20% | -17% | 0.78 |
| CPU_ACTIVITY | -38% | -1% | -61% | 0.96 | 1.29 | 0.24 | -3% | -16% | 0.78 |
| NAVAL_PROPULSION_PLANT | -36% | -4% | -52% | 0.95 | 5.40 | 1.59 | -215% | -562% | -0.85 |
| NAVAL_PROPULSION_PLANT | -44% | -2% | -78% | 0.95 | 4.14 | 1.22 | -99% | -94% | 0.16 |
| NAVAL_PROPULSION_PLANT | -64% | -1% | -175% | 0.98 | 4.66 | 1.37 | -141% | -163% | 0.50 |
| NAVAL_PROPULSION_PLANT | -35% | **2%** | -67% | 0.92 | 3.53 | 1.04 | -24% | -27% | 0.27 |
| NAVAL_PROPULSION_PLANT | -38% | -2% | -70% | 0.94 | 4.75 | 1.40 | -62% | -68% | -0.21 |
| MIAMI_HOUSING | -32% | -1% | -46% | 0.94 | 1.48 | 0.18 | -21% | -20% | 0.74 |
| MIAMI_HOUSING | -46% | **1%** | -88% | 0.96 | 1.98 | 0.25 | -41% | -48% | 0.53 |
| MIAMI_HOUSING | -25% | -2% | -31% | 0.95 | 1.69 | 0.21 | -18% | -19% | 0.75 |
| MIAMI_HOUSING | -34% | -2% | -49% | 0.96 | 1.82 | 0.23 | -23% | -22% | 0.77 |
| MIAMI_HOUSING | -42% | -2% | -72% | 0.91 | 1.91 | 0.24 | -24% | -32% | 0.57 |
| KIN8NM | -14% | **1%** | -17% | 0.98 | 3.68 | 0.72 | -3% | 0% | 0.43 |
| KIN8NM | -18% | 0% | -22% | 0.97 | 3.67 | 0.72 | -20% | -22% | 0.36 |
| KIN8NM | -17% | **1%** | -22% | 0.96 | 3.56 | 0.70 | -12% | -10% | 0.39 |
| KIN8NM | -24% | 0% | -32% | 0.96 | 3.46 | 0.68 | -26% | -24% | 0.36 |
| KIN8NM | -20% | 0% | -25% | 0.95 | 3.51 | 0.69 | -20% | -19% | 0.31 |
| CONCRETE_COMPRESSIVE_STRENGTH | -10% | **2%** | -13% | 0.95 | 2.80 | 0.57 | -9% | -6% | 0.51 |
| CONCRETE_COMPRESSIVE_STRENGTH | **9%** | -3% | **8%** | 0.95 | 2.68 | 0.54 | -15% | -13% | 0.59 |
| CONCRETE_COMPRESSIVE_STRENGTH | -1% | -4% | **3%** | 0.95 | 2.86 | 0.58 | -20% | -23% | 0.52 |
| CONCRETE_COMPRESSIVE_STRENGTH | **16%** | -5% | **16%** | 0.91 | 2.49 | 0.50 | -10% | -10% | 0.46 |
| CONCRETE_COMPRESSIVE_STRENGTH | **13%** | -3% | **11%** | 0.96 | 2.40 | 0.48 | -5% | -9% | 0.60 |
| CARS | -23% | -3% | -33% | 0.95 | 1.93 | 0.33 | -3% | -5% | 0.75 |
| CARS | -28% | -2% | -43% | 0.97 | 1.94 | 0.33 | -36% | -38% | 0.75 |
| CARS | -27% | -3% | -39% | 0.94 | 2.06 | 0.35 | -34% | -33% | 0.70 |
| CARS | -41% | **6%** | -85% | 0.97 | 2.70 | 0.46 | -32% | -37% | 0.63 |
| CARS | **8%** | -4% | **10%** | 0.90 | 1.68 | 0.28 | -32% | -15% | 0.48 |
| ENERGY_EFFICIENCY | -12% | 0% | -14% | 0.99 | 1.44 | 0.42 | -8% | **1%** | 0.97 |
| ENERGY_EFFICIENCY | -4% | -2% | -2% | 0.97 | 0.69 | 0.20 | **14%** | **18%** | 0.98 |
| ENERGY_EFFICIENCY | -24% | -3% | -37% | 0.94 | 1.23 | 0.36 | **8%** | **2%** | 0.92 |
| ENERGY_EFFICIENCY | **10%** | -3% | -1% | 0.96 | 1.07 | 0.31 | -8% | **6%** | 0.96 |
| ENERGY_EFFICIENCY | -7% | -10% | 0% | 0.89 | 0.84 | 0.24 | -1% | -32% | 0.93 |
| CALIFORNIA_HOUSING | -28% | **1%** | -42% | 0.95 | 3.00 | 0.71 | -8% | -9% | 0.55 |
| CALIFORNIA_HOUSING | -20% | -2% | -25% | 0.97 | 2.85 | 0.68 | -1% | 0% | 0.68 |
| CALIFORNIA_HOUSING | -30% | 0% | -42% | 0.96 | 2.90 | 0.69 | -7% | -3% | 0.63 |
| CALIFORNIA_HOUSING | -16% | -1% | -19% | 0.93 | 2.70 | 0.64 | -4% | -3% | 0.54 |
| CALIFORNIA_HOUSING | -28% | 0% | -40% | 0.96 | 2.83 | 0.67 | -2% | -1% | 0.65 |
| AIRFOIL_SELF_NOISE | -14% | 0% | -16% | 1.00 | 3.46 | 0.74 | -20% | -11% | 0.84 |
| AIRFOIL_SELF_NOISE | **2%** | -6% | **7%** | 0.89 | 2.60 | 0.56 | -22% | -13% | 0.31 |
| AIRFOIL_SELF_NOISE | -12% | -8% | -9% | 0.88 | 2.56 | 0.55 | -23% | -20% | 0.44 |
| AIRFOIL_SELF_NOISE | -13% | -5% | -12% | 0.93 | 2.94 | 0.63 | -9% | -1% | 0.62 |
| AIRFOIL_SELF_NOISE | -5% | **1%** | -8% | 0.94 | 2.77 | 0.59 | -3% | **1%** | 0.54 |
| QSAR_FISH_TOXICITY | -16% | **1%** | -20% | 0.99 | 4.78 | 0.74 | -8% | -9% | 0.44 |
| QSAR_FISH_TOXICITY | -7% | -5% | -7% | 0.92 | 3.31 | 0.51 | -21% | -28% | 0.28 |
| QSAR_FISH_TOXICITY | -5% | -2% | -8% | 0.95 | 3.17 | 0.49 | -6% | -8% | 0.39 |
| QSAR_FISH_TOXICITY | -8% | -5% | -8% | 0.86 | 2.74 | 0.43 | -21% | -17% | 0.28 |
| QSAR_FISH_TOXICITY | -17% | **2%** | -28% | 0.90 | 3.03 | 0.47 | -10% | -9% | 0.32 |

## F  Aggregated results for non-conformalized predictions

Table 12: Test results for all models with complete-raw data at 95% quantiles aggregated over five seeds. For each metric, the mean and standard deviation of the performance across the seeds are separated by ±. Performance over the baseline is highlighted in bold.

| Dataset | Interval Predictions | | | | | Point Predictions | | |
|---|---|---|---|---|---|---|---|---|
| | Relative | | | Absolute | | Relative | | Absolute |
| | ΔCWR | ΔPICP | ΔNMPIW | PICP | NMPIW | ΔRMSE | ΔMAE | $R^2$ |
| **MultiXGBs** | | | | | | | | |
| SPACE_GA | -1%±5% | **6%±2%** | -7%±6% | 0.89±0.01 | 0.20±0.01 | -9%±1% | -10%±2% | 0.60±0.01 |
| CPU_ACTIVITY | **25%±6%** | **9%±1%** | **12%±4%** | 0.89±0.01 | 0.07±0.00 | **21%±1%** | -1%±2% | 0.98±0.00 |
| NAVAL_PROPULSION_PLANT | **156%±13%** | **8%±1%** | **58%±2%** | 0.96±0.00 | 0.11±0.00 | -14%±4% | -12%±2% | 0.99±0.00 |
| MIAMI_HOUSING | **61%±5%** | -7%±2% | **43%±3%** | 0.82±0.01 | 0.06±0.00 | **4%±4%** | **3%±3%** | 0.91±0.01 |
| KIN8NM | **10%±3%** | **5%±2%** | **4%±4%** | 0.90±0.01 | 0.38±0.01 | -20%±1% | -22%±1% | 0.63±0.01 |
| CONCRETE_COMPRESSIVE_STRENGTH | **13%±4%** | **26%±6%** | -11%±7% | 0.91±0.01 | 0.25±0.00 | -26%±6% | -40%±11% | 0.85±0.01 |
| CARS | **18%±8%** | **15%±4%** | **2%±8%** | 0.89±0.02 | 0.12±0.01 | -3%±3% | -1%±3% | 0.95±0.00 |
| ENERGY_EFFICIENCY | **247%±89%** | **22%±6%** | **60%±17%** | 0.88±0.02 | 0.05±0.01 | -15%±22% | -16%±17% | 1.00±0.00 |
| CALIFORNIA_HOUSING | **31%±5%** | 0%±1% | **23%±4%** | 0.88±0.01 | 0.28±0.00 | -1%±1% | -1%±1% | 0.81±0.00 |
| AIRFOIL_SELF_NOISE | **61%±41%** | **4%±9%** | **31%±16%** | 0.81±0.07 | 0.19±0.04 | -64%±32% | -73%±41% | 0.86±0.05 |
| QSAR_FISH_TOXICITY | **10%±3%** | **11%±3%** | -1%±3% | 0.76±0.01 | 0.23±0.00 | **3%±3%** | **1%±4%** | 0.55±0.02 |
| **MultiETs** | | | | | | | | |
| SPACE_GA | **9%±3%** | -2%±1% | **9%±4%** | 0.91±0.01 | 0.22±0.01 | -16%±2% | -18%±3% | 0.54±0.02 |
| CPU_ACTIVITY | **27%±3%** | -8%±0% | **27%±2%** | 0.90±0.00 | 0.09±0.00 | -14%±1% | -20%±2% | 0.98±0.00 |
| NAVAL_PROPULSION_PLANT | **219%±32%** | -9%±0% | **71%±3%** | 0.91±0.00 | 0.19±0.02 | -320%±41% | -409%±36% | 0.96±0.01 |
| MIAMI_HOUSING | **69%±3%** | -13%±0% | **48%±1%** | 0.86±0.00 | 0.08±0.00 | -10%±2% | -20%±1% | 0.90±0.00 |
| KIN8NM | **10%±2%** | -9%±0% | **17%±2%** | 0.88±0.01 | 0.44±0.01 | -35%±2% | -38%±2% | 0.48±0.01 |
| CONCRETE_COMPRESSIVE_STRENGTH | **9%±9%** | -12%±2% | **19%±8%** | 0.81±0.01 | 0.24±0.02 | -67%±7% | -94%±9% | 0.76±0.01 |
| CARS | **11%±6%** | **16%±2%** | -5%±7% | 0.83±0.02 | 0.09±0.01 | **3%±3%** | **1%±2%** | 0.95±0.00 |
| ENERGY_EFFICIENCY | -4%±5% | 0%±7% | -5%±14% | 0.65±0.03 | 0.02±0.00 | **3%±3%** | 0%±1% | 1.00±0.00 |
| CALIFORNIA_HOUSING | **31%±2%** | -7%±0% | **29%±1%** | 0.90±0.00 | 0.41±0.01 | -15%±1% | -21%±1% | 0.71±0.01 |
| AIRFOIL_SELF_NOISE | **18%±25%** | -15%±4% | **25%±17%** | 0.83±0.04 | 0.26±0.06 | -117%±45% | -141%±51% | 0.80±0.08 |
| QSAR_FISH_TOXICITY | **12%±2%** | -9%±1% | **19%±2%** | 0.82±0.01 | 0.28±0.00 | -6%±1% | -11%±1% | 0.53±0.01 |
| **MultiMLPs** | | | | | | | | |
| SPACE_GA | **8%±1%** | 0%±3% | **7%±3%** | 0.81±0.01 | 0.14±0.00 | 0%±1% | -1%±1% | 0.75±0.01 |
| CPU_ACTIVITY | **8%±3%** | -9%±4% | **15%±6%** | 0.72±0.02 | 0.05±0.00 | **7%±2%** | **5%±2%** | 0.98±0.00 |
| NAVAL_PROPULSION_PLANT | **21%±18%** | 0%±3% | **16%±13%** | 0.92±0.01 | 0.07±0.01 | -45%±25% | -36%±20% | 1.00±0.00 |
| MIAMI_HOUSING | **9%±4%** | **2%±3%** | **6%±6%** | 0.81±0.02 | 0.05±0.00 | **4%±2%** | **4%±2%** | 0.91±0.00 |
| KIN8NM | **2%±7%** | **8%±5%** | -6%±10% | 0.81±0.02 | 0.13±0.01 | **7%±2%** | **7%±2%** | 0.93±0.00 |
| CONCRETE_COMPRESSIVE_STRENGTH | **98%±13%** | -23%±7% | **61%±5%** | 0.56±0.04 | 0.06±0.00 | **12%±5%** | **15%±4%** | 0.91±0.01 |
| CARS | 0%±14% | **6%±12%** | -9%±30% | 0.76±0.08 | 0.07±0.01 | -3%±3% | 0%±3% | 0.95±0.00 |
| ENERGY_EFFICIENCY | **82%±18%** | **25%±23%** | **31%±11%** | 0.62±0.04 | 0.02±0.00 | **32%±3%** | **33%±3%** | 1.00±0.00 |
| CALIFORNIA_HOUSING | -14%±2% | **8%±1%** | -26%±3% | 0.89±0.01 | 0.31±0.01 | **7%±0%** | **9%±1%** | 0.82±0.00 |
| AIRFOIL_SELF_NOISE | **74%±41%** | 0%±6% | **39%±15%** | 0.75±0.02 | 0.08±0.00 | **18%±4%** | **16%±5%** | 0.97±0.00 |
| QSAR_FISH_TOXICITY | -6%±4% | **11%±3%** | -19%±5% | 0.76±0.03 | 0.23±0.01 | **4%±5%** | **7%±4%** | 0.55±0.04 |

Table 13: Test results for all models with missing-raw data at 95% quantiles aggregated over five seeds. For each metric, the mean and standard deviation of the performance across the seeds are separated by ±. Performance over the baseline is highlighted in bold.

| | Interval Predictions | | | | | Point Predictions | | |
| | Relative | | | Absolute | | Relative | | Absolute |
| Dataset | ΔCWR | ΔPICP | ΔNMPIW | PICP | NMPIW | ΔRMSE | ΔMAE | $R^2$ |
|---|---|---|---|---|---|---|---|---|
| **MultiXGBs** | | | | | | | | |
| SPACE_GA | -5%±6% | **4%±2%** | -12%±7% | 0.86±0.02 | 0.22±0.01 | -7%±4% | -9%±3% | 0.45±0.06 |
| CPU_ACTIVITY | -5%±19% | **12%±8%** | -37%±46% | 0.89±0.06 | 0.14±0.04 | -1%±21% | -17%±13% | 0.83±0.07 |
| NAVAL_PROPULSION_PLANT | **105%±29%** | -16%±6% | **56%±6%** | 0.70±0.06 | 0.18±0.02 | -14%±20% | -29%±35% | 0.77±0.05 |
| MIAMI_HOUSING | **16%±13%** | -5%±5% | **10%±16%** | 0.82±0.06 | 0.10±0.02 | -14%±11% | -18%±10% | 0.72±0.10 |
| KIN8NM | -5%±2% | **3%±2%** | -14%±5% | 0.88±0.01 | 0.48±0.01 | -3%±3% | -4%±4% | 0.40±0.02 |
| CONCRETE_COMPRESSIVE_STRENGTH | **2%±3%** | **8%±4%** | -10%±3% | 0.77±0.03 | 0.30±0.01 | -16%±12% | -20%±14% | 0.58±0.08 |
| CARS | -15%±14% | **26%±9%** | -63%±26% | 0.85±0.04 | 0.27±0.03 | -30%±22% | -35%±24% | 0.66±0.15 |
| ENERGY_EFFICIENCY | **10%±26%** | **11%±6%** | -24%±43% | 0.92±0.02 | 0.22±0.05 | -186%±265% | -153%±188% | 0.91±0.08 |
| CALIFORNIA_HOUSING | **26%±7%** | -2%±3% | **21%±6%** | 0.88±0.03 | 0.40±0.03 | -4%±4% | -5%±5% | 0.69±0.05 |
| AIRFOIL_SELF_NOISE | -2%±18% | -3%±4% | -4%±17% | 0.73±0.04 | 0.34±0.06 | -37%±31% | -43%±36% | 0.41±0.12 |
| QSAR_FISH_TOXICITY | -2%±3% | 0%±5% | -6%±7% | 0.73±0.03 | 0.26±0.02 | -4%±4% | -2%±3% | 0.36±0.04 |
| **MultiETs** | | | | | | | | |
| SPACE_GA | **4%±3%** | -5%±1% | **7%±4%** | 0.88±0.02 | 0.24±0.01 | -12%±4% | -14%±4% | 0.40±0.05 |
| CPU_ACTIVITY | -16%±10% | -5%±1% | -20%±18% | 0.93±0.01 | 0.20±0.03 | -30%±23% | -54%±20% | 0.79±0.10 |
| NAVAL_PROPULSION_PLANT | **58%±5%** | -17%±4% | **46%±3%** | 0.82±0.05 | 0.41±0.03 | -36%±21% | -93%±48% | 0.71±0.05 |
| MIAMI_HOUSING | **20%±14%** | -10%±4% | **21%±13%** | 0.87±0.05 | 0.14±0.03 | -27%±11% | -37%±11% | 0.67±0.11 |
| KIN8NM | **5%±2%** | -8%±2% | **12%±4%** | 0.87±0.02 | 0.50±0.02 | -11%±2% | -13%±3% | 0.32±0.03 |
| CONCRETE_COMPRESSIVE_STRENGTH | **1%±5%** | -14%±3% | **13%±2%** | 0.80±0.02 | 0.37±0.01 | -35%±21% | -50%±25% | 0.49±0.06 |
| CARS | -23%±6% | -8%±5% | -35%±14% | 0.83±0.04 | 0.24±0.02 | -33%±18% | -31%±18% | 0.66±0.09 |
| ENERGY_EFFICIENCY | -15%±16% | 0%±9% | -36%±42% | 0.85±0.05 | 0.15±0.01 | -73%±74% | -84%±64% | 0.94±0.04 |
| CALIFORNIA_HOUSING | **21%±7%** | -7%±2% | **22%±5%** | 0.91±0.02 | 0.52±0.03 | -14%±3% | -17%±3% | 0.61±0.05 |
| AIRFOIL_SELF_NOISE | -7%±18% | -20%±2% | **8%±16%** | 0.76±0.03 | 0.42±0.08 | -65%±54% | -93%±70% | 0.33±0.11 |
| QSAR_FISH_TOXICITY | **10%±7%** | -9%±3% | **16%±3%** | 0.82±0.03 | 0.33±0.03 | -8%±8% | -10%±9% | 0.39±0.09 |
| **MultiMLPs** | | | | | | | | |
| SPACE_GA | -11%±5% | -15%±2% | **1%±6%** | 0.68±0.03 | 0.16±0.01 | -40%±23% | -34%±15% | 0.21±0.23 |
| CPU_ACTIVITY | -50%±16% | **22%±14%** | -193%±118% | 0.86±0.10 | 0.18±0.07 | -15%±19% | -23%±17% | 0.75±0.12 |
| NAVAL_PROPULSION_PLANT | -52%±19% | -14%±24% | -102%±56% | 0.63±0.18 | 0.34±0.07 | -108%±66% | -183%±195% | -0.03±0.47 |
| MIAMI_HOUSING | -33%±15% | **13%±10%** | -92%±70% | 0.85±0.06 | 0.12±0.04 | -25%±8% | -28%±11% | 0.67±0.10 |
| KIN8NM | -11%±5% | -5%±2% | -8%±6% | 0.77±0.01 | 0.35±0.02 | -16%±8% | -15%±9% | 0.37±0.04 |
| CONCRETE_COMPRESSIVE_STRENGTH | -15%±16% | -13%±28% | -10%±48% | 0.60±0.22 | 0.22±0.11 | -12%±5% | -12%±6% | 0.54±0.05 |
| CARS | -51%±4% | **43%±7%** | -211%±34% | 0.93±0.02 | 0.29±0.03 | -28%±12% | -26%±13% | 0.66±0.10 |
| ENERGY_EFFICIENCY | -13%±18% | **9%±15%** | -30%±25% | 0.66±0.13 | 0.09±0.04 | **1%±9%** | -1%±16% | 0.95±0.03 |
| CALIFORNIA_HOUSING | -17%±6% | **4%±3%** | -28%±14% | 0.88±0.03 | 0.44±0.04 | -5%±2% | -3%±3% | 0.61±0.06 |
| AIRFOIL_SELF_NOISE | **2%±16%** | -2%±5% | -1%±21% | 0.75±0.07 | 0.25±0.08 | -15%±8% | -9%±8% | 0.55±0.18 |
| QSAR_FISH_TOXICITY | -17%±5% | **6%±6%** | -30%±11% | 0.76±0.04 | 0.28±0.03 | -13%±7% | -14%±8% | 0.34±0.06 |

## G  EXAMPLE OF LEARNING WITH NON-NORMALITY

To illustrate how SEMF adapts to non-normal outcomes, we provide an example from the *naval_propulsion_plant dataset* (Coraddu et al., 2016). Figure 2 shows the distribution of the ground-truth $y$ variable which in this case is `gt_compressor_decay_state_coefficient`. The values are uniformly distributed, and we only standardize the values without changing the shape of the distribution.

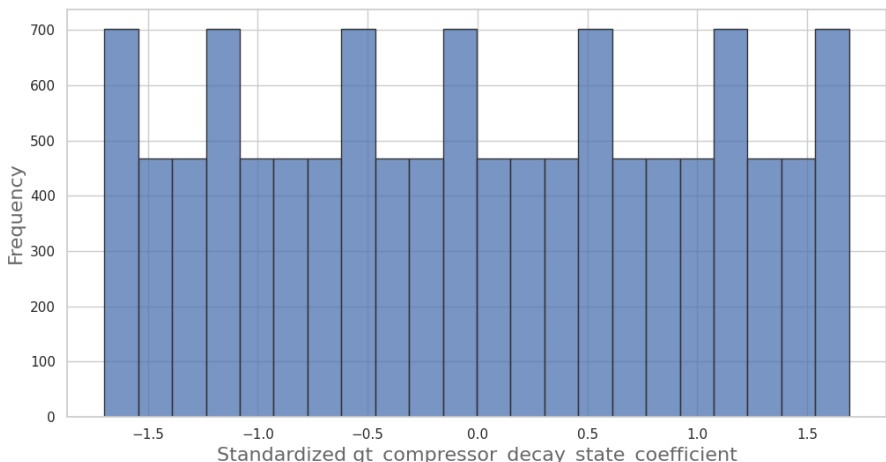

Figure 2: Distribution of the standardized outcome ($y$) variable for the `naval_propulsion_plant` dataset which shows that $y$ is uniformly distributed prior to any training.

After training our SEMF model under the normality assumption with the ideal hyper-parameters (and a seed of 0), sampling from a normal distribution for the $z$ dimension, we infer on some randomly sampled test instances that provide us with the prediction intervals in Figure 3. The 'SEMF intervals' can be compared with XGBoost quantile regression, constituting our baseline. This figure shows that SEMF's predicted intervals are better than the baseline. This plot alone does not tell us much about the predicted output distribution. Therefore, we provide Figure 4. The last plot shows that for a handful of the instances, the predicted values can take any shape and are not necessarily normal.

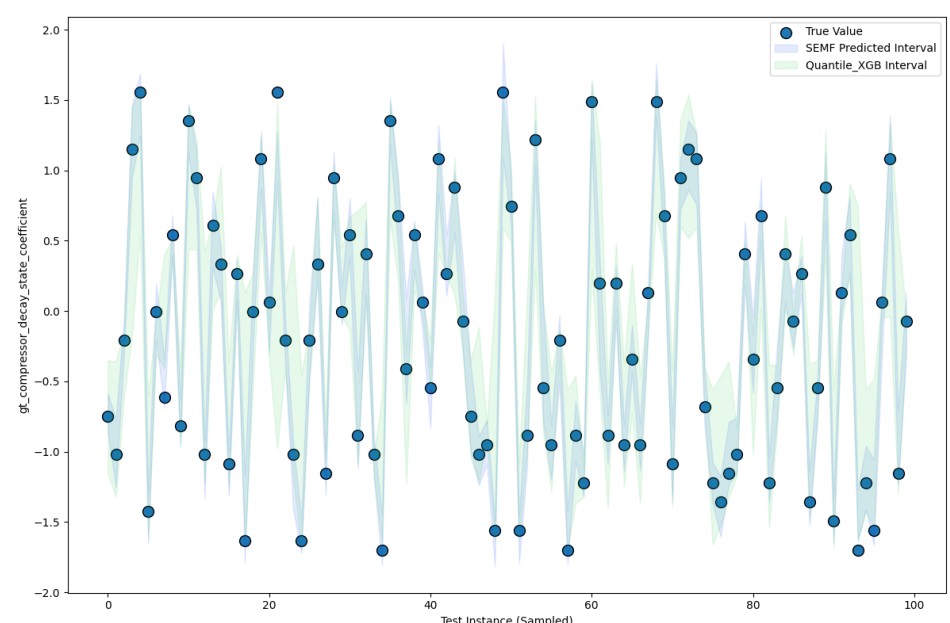

Figure 3: Predicted intervals and true values on 100 randomly selected test samples for SEMF MultiXGBs and XGBoost quantile regression with complete data at 95% quantiles (according to Eq 13).

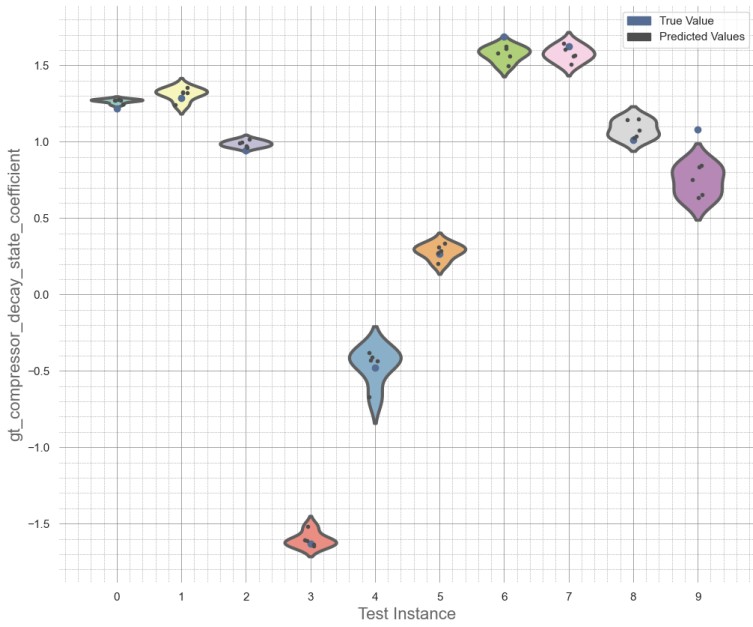

Figure 4: Violin plot for the first ten instances of the test set where both the ground truth and 50 values inferred by SEMF (according to Eq 12) are generated for each instance. We have added jitter here, so the points do not perfectly align along the x-axis.

