# OpenReview forum: "SEMF: Supervised Expectation-Maximization Framework for Predicting Intervals"
_ICLR.cc/2025/Conference — ICLR 2025 Conference Withdrawn Submission_

### Official Review · Reviewer_1Q7G · 2024-10-24

**Soundness:** 3
**Presentation:** 3
**Contribution:** 2
**Rating:** 3
**Confidence:** 3

**Summary:**

This paper proposes an Expectation Maximization (EM) Approach for supervised learning problems. The algorithm is described and evaluated empirically on several benchmark models and datasets. I am not yet convinced by the approach but happy to change my mind during the discussion period.

**Strengths:**

- The approach model agnostic and so can in theory easily applied on any supervised learning baseline model.
- The overall algorithm is well described and easy to follow, at least for someone who has worked on EM algorithms.

**Weaknesses:**

- Using MC sampling to approximate the posterior over the latent variables z is a potentially inaccurate approach, especially as z increases in dimensionality. This is the reason, in Bayesian inference, we would e.g. use MCMC sampling not MC sampling from the prior. Can the authors comment on why this is not a problem in their approach? Asked differently, how large did the number of samples R have to be in their cases to produce good results? How does R affect the quality of the results?
- The results not super convincing across the board. Especially, I was surprised to see the point estimation performance to drop when using the EM approach. Can the authors explain that?
- The missing value setting is strange to me. Why investigate an MCAR setting, which implies ignorable missingness. I believe a MAR setting where missingness is in principle recoverable would be a much more relevant scenario.

**Questions:**

- Why do we need the double index r, s when sampling first z and then y in EQ 17?  Just sampling z_r and then y_r would be conceptually enough I believe.
- Why does mini-batching make your results unstable? Is that a common reason or something specific to your method?

---

> ### Author Response · Authors · 2024-11-21
>
> We thank reviewer `1Q7G` for their thoughtful review and for raising important points.
>
> > Using MC sampling to approximate the posterior over the latent variables z is a potentially inaccurate approach... in Bayesian inference, we would e.g. use MCMC sampling not MC sampling...
>
> We highly appreciate this comment by the reviewer as it is an excellent suggestion, albeit it falls outside the scope of this paper. Since we work with normal distributions in both the encoder and decoder (Section 2.2.1), direct sampling is both efficient and sufficient—the computational overhead of MCMC may not provide additional benefits. Our experiments demonstrate that relatively few samples (R=25-50) achieve good performance, with diminishing returns beyond 50 samples. This is evidenced by the strong performance of XGBoost on naval\_propulsion\_plant and energy\_efficiency datasets, where SEMF achieved 172\%±14\% and 222\%±45\% improvements in $\Delta$CWR, respectively. While higher dimensions could theoretically impact sampling efficiency, our empirical results across datasets with varying dimensionality (features in [7,22] and $m_k$ up to 30 for each feature) show that simple MC sampling remains effective within these ranges when working with normal distributions. Nevertheless, MCMC remains an interesting venue for follow-up works, especially for studying $\mathcal{L}(\phi, \theta, \xi)$ under much more complex distributions.
>
> > The results not super convincing across the board. Especially, I was surprised to see the point estimation performance to drop when using the EM approach. Can the authors explain that?
>
> The apparent decrease in point estimation performance is an expected trade-off given our framework's primary focus on interval prediction. SEMF introduces stochasticity through sampling operations, which, while beneficial for capturing uncertainty and generating robust prediction intervals, can impact point predictions. As lines 319-320 explained, we targeted $\sigma_k$ that introduces more noise and produces better intervals than point predictions. The idea was to widen the latent space, indirectly resulting in wider prediction intervals for the output.
> We observe that for neural networks, where in lines 429-467, we explain how MultiMLPs, in some cases, did not suffer from the same issue since the injected noise was helpful to avoid overfitting but not high enough to create wide intervals.
>
> > The missing value setting is strange to me. Why investigate an MCAR setting, which implies ignorable missingness. I believe a MAR setting where missingness is in principle recoverable would be a much more relevant scenario.
>
> We agree that MAR scenarios would provide additional insights. Our choice of MCAR was primarily motivated by two factors: First, it provides a clear baseline for evaluating our framework's basic capability to handle missing data without conflating it with the complexity of missingness mechanisms. Second, it allows for direct comparison with existing imputation methods (particularly relevant for mean and median imputations). We acknowledge this limitation in Section 7 and agree that extending to MAR scenarios would be valuable for future work.
>
> > Why do we need the double index r, s when sampling first z and then y in EQ 17?  Just sampling z\_r and then y\_r would be conceptually enough I believe.
>
> The double indexing (r, s) in equation 17 serves a specific purpose: it allows us to generate multiple y predictions for each z sample, providing a richer characterization of the prediction distribution. This is particularly important when the decoder's uncertainty differs from the uncertainty in the latent space. However, we acknowledge that for simpler applications, using a single index as suggested could be sufficient.
>
> > Why does mini-batching make your results unstable? Is that a common reason or something specific to your method?
>
> Regarding the theory, there is no reason for mini-batching to make the results unstable. The reviewer may be referring to line 303, where we stated, *''all data in SEMF are processed batch-wise, without employing mini-batch training, to ensure consistency and stability in the training process''*. In line 181 (theoretical perspective), we explain that mini-batches can be used; however, in this simple case where we treat low-dimensional tabular data, it is unnecessary since everything can fit into memory. With high-dimensional data, mini-batching may be needed. In line 181, we intended to convey that we do not use mini-batching so that both SEMF and the baselines train on the same subset of the data to avoid an unfair comparison. We do not know the exact impact of mini-batching within SEMF (not referring to its individual $p_\phi$ and $p_\theta$ models); therefore, to avoid confusion, we will remove the word 'stability' from line 303.
>
> We thank the reviewer for the intriguing discussion and for reading our work thoroughly.

---

### Official Review · Reviewer_uZyy · 2024-11-02

**Soundness:** 2
**Presentation:** 1
**Contribution:** 2
**Rating:** 3
**Confidence:** 3

**Summary:**

This paper presents a novel prediction interval generation framework called SEMF (Supervised Expectation-Maximization Framework). SEMF is a general, model-agnostic approach that can be applied to complete datasets or datasets containing missing data. It extends the traditional EM (Expectation-Maximization) algorithm, which is typically used for unsupervised learning, by applying it to supervised learning for uncertainty estimation through latent variable modeling.

**Strengths:**

- The SEMF proposed by the authors is a model-agnostic method, meaning it can be integrated with various machine learning models, providing high applicability and flexibility.
- The problem addressed by the authors is often overlooked in the real world, specifically the presence of missing data and the uncertainty estimation of provided predictions.
- The authors conducted experiments on a large number of datasets to validate the effectiveness of their method.

**Weaknesses:**

- The writing in this paper is unclear. I suggest that the authors introduce the research problem setting either before Section 2 or at the beginning of Section 2, rather than listing formulas.
- The theoretical analysis in the paper assumes independent distributions among the variables, but real-world situations are often more complex. I believe the authors' investigation of this issue is not sufficiently thorough.
- There appears to be a substantial amount of prior research [1, 2, 3] on interval data prediction, which is not mentioned in this paper.

[1] Billard, L. and Diday, E. Regression analysis for interval valued data. In Data Analysis, Classification, and Related Methods, pp. 369–374. Springer, 2000.


[2] Sadeghi, J., De Angelis, M., and Patelli, E. Efficient training of interval neural networks for imprecise training data. Neural Networks, 118:338–351, 2019.


[3] Yang, Z., Lin, D. K., and Zhang, A. Interval-valued data prediction via regularized artificial neural network. Neurocomputing, 331:336–345, 2019.

**Questions:**

- The authors assume independent distributions among the variables. Based on this assumption, is the hypothesis of the latent variable $z$ not that important? Could we establish the relationship between
$y$ and $x$ directly instead?
- There is a significant amount of research on interval prediction (which the authors also mention), but the authors do not seem to compare their method with these existing approaches (in the absence of missing data). For cases with missing data, using simple methods (such as interpolation) for comparison would also be straightforward.
- The SEMF method seems to require tuning multiple hyperparameters, such as the number of Monte Carlo samples, the number of latent nodes, and the standard deviation. This may demand considerable experimental and computational resources. Are there any general empirical settings that could be recommended?

---

> ### Author Response · Authors · 2024-11-21
>
> We appreciate the feedback from reviewer `uZyy`, and will address their concerns.
>
> > The writing in this paper is unclear. I suggest that the authors introduce the research problem setting either before Section 2 or at the beginning of Section 2, rather than listing formulas.
>
> We are grateful for the suggestion regarding the writing, but the mathematical formulations are necessary to understand the framework's underlying foundations. The first exposure to the formulas is in Equation 1, where we explain what each variable represents right below it (line 065). Then, we specify the notations for our work in Section 2.1, lines 070-084. Given that the other reviewers pointed out that the paper is easy to follow, we would happily consider any concrete suggestions the reviewer can share to make necessary modifications.
>
> > The theoretical analysis in the paper assumes independent distributions among the variables, but real-world situations are often more complex. I believe the authors' investigation of this issue is not sufficiently thorough.
>
> There may have been a misunderstanding. We do not assume 'independent distribution of the variables' but rather i.i.d. data, meaning that each row is independent of one another (that is, we are not dealing with time series). This is a common assumption behind many statistical models and even the most prominent interval prediction approaches, such as conformal prediction. The EM algorithm also entails the use of i.i.d. data. Adapting our work for time series is possible but falls beyond the current scope of the paper, which aims to introduce the underlying foundations.
>
> > The authors assume independent distributions among the variables. Based on this assumption, is the hypothesis of the latent variable $z$ not that important? Could we establish the relationship between $y$ and $x$ directly instead?
>
> We have partially addressed this comment above, but we would like to provide further explanations. Lines 078-81 mention that inputs to our framework are 'independent conditionally on their corresponding source'. This means that each input $x$, for instance, $x_1$, gets its own set of $z_1$ before the fusion (concatenation), with $z_2$ coming from $x_2$. Establishing a relationship between $y$ and $x$ defeats the whole purpose of a probabilistic modeling approach, and it is indeed what our baselines for point prediction already do.
>
> > There is a significant amount of research on interval prediction (which the authors also mention)...
>
> Thank you for bringing these references to our attention. While the cited works make valuable contributions, they differ from SEMF in key aspects. *Billard & Diday (2000)* focuses on regression with interval-valued inputs rather than generating prediction intervals. As a disclaimer, we could not access the full paper as it is behind a paywall, but accessing the first two pages made it clear that it addresses a different problem than the one we consider. *Sadeghi et al. (2019)* and *Yang et al. (2019)* propose model-specific approaches for neural networks, whereas SEMF is model-agnostic. Hence, the cited references are not directly comparable to SEMF, a **model-agnostic** framework that produces prediction **intervals**. Nonetheless, we are open to reconsidering if the reviewer can point out specific aspects that make them relevant to our work.
>
> > The SEMF method seems to require tuning multiple hyperparameters...Are there any general empirical settings that could be recommended?
>
> We thank the reviewer for raising this interesting point. This valid criticism generally applies to all the underlying machine learning models we have used (especially neural networks) on top of our framework. Based on our experiments, we can happily recommend some practical hyperparameter settings:
>
> - **Monte Carlo samples (R):** 25-50 samples typically balance accuracy and computational cost well. Going beyond 50 samples yields diminishing returns. Note that one can set a smaller number of samples during training and a higher one during inference.
>
> - **Latent nodes ($m_k$):** For most tabular datasets, 10-20 nodes per encoder are sufficient, but again, this is under the setting of using one input per encoder, where in reality, the user can group their relevant input features. Higher R alongside higher $m_k$ can result in better models at the cost of longer computation.
>
> - **Standard deviation ($\sigma_k$):** Values between 0.1-1.0 work well for standardized inputs. Smaller values (0.01-0.1) produce narrower intervals but may underestimate uncertainty, while larger values (>1.0) tend to overestimate it.
>
> Additionally, early stopping with patience of 5-10 steps helps reduce computational overhead without sacrificing performance. These are our recommendations based on the 330 experiments of the three models (MultiXGBs, MultiETs, MultiMLPs) across the 11 datasets.
>
> We thank the reviewer again for their time and feedback.

---

### Official Review · Reviewer_Y2Ma · 2024-11-03

**Soundness:** 3
**Presentation:** 3
**Contribution:** 2
**Rating:** 3
**Confidence:** 3

**Summary:**

This paper introduces an algorithm for producing prediction intervals (PI) by adapting the EM algorithm in a supervised learning framework. The modeling takes into consideration possible missing input features. The empirical evaluation considers the 95% PI (its coverage and width) against baseline models.

**Strengths:**

- Adapting the EM algorithm seems pretty novel and the algorithm is well-motivated and sound.
- There have been many methods introduced in the uncertainty literature which aim to produce prediction intervals, and this work would fit alongside those methods.
- Consideration for missing data is an interesting application setting which prior works in uncertainty to not consider often.
- Overall, the writing quality is good and mostly easy to follow

**Weaknesses:**

- The proposed method incorporates conformal prediction at the end, which guarantees correct coverage. the only other metric is the PI width. In that case, is the benefit of the method just in producing more tightly clustered samples which lead to tighter PI?
- Is it correct to understand the proposed method as just producing samples from the modeled underlying distribution? There are other metrics which are sample-based that could provide a more holistic evaluation of the quality of the predictive distribution, like the energy score.
- As I understand it, this method requires a separate model per input feature dimension, which seems prohibitively expensive, either for large models or large input dimensions.
- Despite the emphasis on adaptability to missing data in the methods sections, its performance on missing data seems a bit unfortunate, but I appreciate the authors' frankness in providing the results.

**Questions:**

- In L153, why is the $x_1[j]$ notation needed? Isn't this already expressed as $x_{1,j}$?
- In equation 12, what is $x_{1}^{(nm)}[j]$? I don't think $x_{1}^{(nm)}$ is defined.
- I don't quite understand how the missing values were implemented - in L299, when you say "except for the first feature", does that mean the first feature is never missing, and half of the rest of the features are masked out, chosen at random?

---

> ### Author Response · Authors · 2024-11-21
>
> We thank reviewer `Y2Ma` for their feedback and will happily address their questions and concerns.
>
> > The proposed method incorporates conformal prediction at the end...
>
> The method does indeed produce tighter intervals while achieving similar coverage, which is a sign that the overall interval quality improves compared to the baseline. The approach is model-agnostic and allows the user to combine different kinds of models (for example, an XGBoost for the first encoder and a neural network for the second encoder), which is particularly beneficial for combining latent representations from multi-modal sources (line 041). Our paper lays the theoretical foundation of such applications for future multi-modal exploration (line 518).
>
> > Is it correct to understand the proposed method as just producing samples from the modeled underlying distribution? There are other metrics...like the energy score.
>
> We thank the reviewer for this question and will do our best to respond despite our uncertainty about what the reviewer meant by *``just producing samples from the modeled underlying distribution''*. As mentioned in Section 2.2.1, the normal distribution, which likely corresponds to the 'underlying distribution' mentioned by the reviewer, is only assumed for simplicity (lines 121-123). As stated in line 507 and Appendix G, even under such normality assumption, this does not entail the predicted intervals to have the same properties, and multi-modal prediction intervals can be generated. We are unaware of energy scores for prediction intervals and would gladly consider any available references the reviewer finds relevant.
>
> > As I understand it, this method requires a separate model per input feature dimension...
>
> The reviewer raises a valid concern about scalability. To clarify, while we presented SEMF with separate encoders per feature for theoretical clarity, in practice, features can be grouped based on domain knowledge or computational constraints. At minimum, the user must include two encoder $p_\phi$ and one decoder $p_\theta$ models. Features that constitute a 'modality' for the user can be grouped to reduce computational overhead. For simplicity, we have one $p_\phi$ per feature, the most computationally expensive way of using the framework. Evidently, due to the re-training of the model and the sampling operation, our approach will always be more computationally expensive than a single model to predict $y$ from $x$ directly.
>
> > Despite the emphasis on adaptability to missing data...
>
> We appreciate the reviewer's comment, and similar to reporting the positive results from the complete data, we sought to transparently show the results from the current state of handling the missing inputs. The relative performance for the missing data is compared to the best possible baseline for that particular metric. We believe this is due to the missingness module design (missing data simulator) and the same hyperparameters found on the complete data to simplify the evaluations (line 486). As mentioned in line 477, since we observe that the approach deals well with complete data, in practice, one can decide on the best imputation method based on the validation data and then impute it using that technique before providing it to SEMF as if the data were complete. Given the results of the complete data, we believe that our ablation study would perform well; however, from a research perspective, it is worth exploring ways to optimize the missing value handling better.
>
> > In L153, why is the $x_1[j]$ notation needed? Isn't this already expressed as $x_{1,j}$?
>
> Thank you for raising this point. The notation $x_1[j]$ and $x_{1,j}$ are slightly different. $x_1[j]$ refers to selecting/indexing the j-th instance of the complete subset of the training data, $I_{nm}$ (for which we also have all the $x_2$ values present), allowing us to substitute the missing $x_1$ with the $x_1$ from $I_{nm}$. $x_{1,j}$ refers to any $x_1$ that is non-missing without this notion of selecting the row $j$.
>
> > In equation 12, what is $x_{1}^{(nm)}[j]$? I don't think $x_{1}^{(nm)}$ is defined.
>
> It is true that since in equation 12 we mention $j\in I_{nm}$, the use of $x_1^{(nm)}[j]$ is unnecessary. We only maintained it in the indicator function to clarify that there is a procedure where we first select the row index, then take the missing value from it (that is, $x_1^{(nm)}[j]$), where we then assign it to the missing ($x_1$). Keeping it as $\mathbf {1}\\{x_1 = x_1[j]\\}$ may also cause slight confusion, but we can maintain this version for the reviewer's clarity.
>
> > I don't quite understand how the missing values were implemented - in L299...
>
> The reviewer's understanding is correct. The first feature is always complete (can be seen as $x_2$ in the simple setup), while the other inputs have been randomly removed at 50\% (can be seen as a subset of $x_1$).
>
> Again, we thank the reviewer for raising interesting points and for their valuable feedback.

---

### Official Review · Reviewer_nPQN · 2024-11-03

**Soundness:** 2
**Presentation:** 3
**Contribution:** 2
**Rating:** 3
**Confidence:** 3

**Summary:**

Quantification of uncertainty of a given prediction made by the models is critical for a variety of downstream applications. One way to measure the uncertainty of a prediction is to use quantile estimates and corresponding prediction intervals. The paper develops and explores SEMF: a Semi-supervised EM Framework for generating prediction intervals that are model agnostic and can work with incomplete data. The gist of the framework is to convert inputs to a latent space, that is in turn used to predict the outputs, much like autoencoder architectures. The proposed interval estimation framework is tested on 11 problems and three different baseline prediction models.

**Strengths:**

Importance: Developing methods and models for improving predictions with their uncertainty estimates is significant and important for applications of predicted models especially when incorrect predictions may carry a high risk. Model agnostic framework the work aims to develop would be a great advantage as it does not have to be tailored to specifics of individual models.

Novelty: The proposed EM framework is novel in context of uncertainty assessment. The methodology of EM and its use in unsupervised and semi-supervised tasks settings is not new.  In addition, the paper introduces a new evaluation metric Coverage-Width Ratio (CWR) that accounts for both the coverage and the precision of the prediction intervals.

Experiments: the experiments are done on 11 problems. Three different prediction methods, and their existing prediction interval estimates are considered. However, the number of baseline models considered does not appear to be sufficient to support the model-agnostic objective.

Code: Authors provide the code for the reproducibility of the reported results

**Weaknesses:**

[W1] Intuition. Lack of intuition and argument supporting the benefit of the SEMF framework. The paper says it aims to leverage latent variable modeling framework for uncertainty estimation in predictions. The intuition and justification of this step and design is however not very well argued in the paper. Adding the text covering the intuition aspect would greatly enhance the readability and clarity of the paper and its steps. Along the same lines it would be great to see arguments or intuition why this approach could improve upon alternative quantile estimation methods.


[W2] Experiment – baselines. The evaluation is limited as it only considers quantile regression as the baseline of comparison that generates prediction interval. Perhaps [1,2], or other relevant methods can be considered as additional baselines. Also, the evaluation is presented on relatively simple datasets: number of features in [7,22] and number of samples in [768, 21K]. It is hard to justify the applicability of the model at scale i.e., with a greater number of features and/or number of samples.

[W3] Experiment- metrics. The results for interval estimates consider PICP, PIW (NPIW) and new CWR metrics, where CWR attempts to combine two different aspects of the interval estimates,  However, there is another existing combined interval score (Gneiting , 2007) that attempts to combine two aspects of the interval estimates and could have been used instead
T. Gneiting, F. Balabdaoui, AE Raftery. Probabilistic forecasts, calibration and sharpness. Journal of the Royal Statistical Society: Series B (Statistical Methodology), 69 (2):243–268, 2007

[W4] Interpretation of results and conclusions: The results and their discussion are limited, and it is unclear whether the objectives of the development are supported by the experiments and results.  First, using only three model baselines is limiting in terms of support of the model agnostic objective. Second, somewhat surprising are missing data experiments where latent variables in the model should adapt and handle better the data missingness.  What is the insight for these results?

[W5] Running time: As authors note in the limitations, the usefulness of this approach is limited by its high computation complexity. However, the paper does not report the train and inference time. For completeness, the paper can benefit from including these times to assess the extent of the slowdown caused by SEMF.

**Questions:**

[W2] Can you elaborate on why you have decided to propose a new score CWR instead of using the existing interval score as suggested in the above comments?

[W4] Can you please comment on why the results of SEMF on missing data are inferior to results on complete data when compared to baseline methods.

---

> ### Author Response · Authors · 2024-11-21
>
> We thank reviewer `nPQN` for their feedback and will happily address all your points and concerns chronologically. Due to the one rebuttal limit per review of 5000 characters, we have reposted truncated versions of the reviewer's original points.
>
> > [W1] Intuition
>
> We appreciate the request for better intuition. SEMF leverages latent variable modeling for two key reasons: First, the latent space provides a natural mechanism for handling missing data through its probabilistic framework. Second, by modeling uncertainty in this latent space and propagating it through the decoder, we can generate prediction intervals without requiring model-specific modifications. The EM algorithm's iterative nature helps refine both the latent representations and their uncertainties, leading to better-calibrated intervals.
>
> > [W2] Experiment – baselines
>
> We would appreciate it if the reviewer could share the references above. In terms of the baselines, quantile regression, in a similar spirit to conformal prediction, has been extended to most models; hence, to the best of our knowledge, it is the only true 'model-agnostic' approach that can be compared to ours, but we would happily look at any references the reviewer provides. We have already included a shorter part about scalability in the limitations in lines 510-511. We want to clarify that any number of features can be given to an encoder $p_\phi$ instead of building multiple ones; here, for simplicity, we have treated each feature as a separate input, but in practice, the user can use as many features as they see fit for a single model. The paper presents the most drastic case of using the framework (one $p_\theta$ model per input) in terms of computational complexity. In contrast to the number of features, observations can increase the overall training time, but since we did not experiment with that, we cannot comment on that aspect.
>
> > [W3] Experiment- metrics
>
> We thank the reviewer for suggesting the Continuous Ranked Probability Score (CRPS). Indeed, CRPS could provide a more comprehensive evaluation of our probabilistic predictions, as it assesses both calibration and sharpness of the entire predictive distribution rather than just interval endpoints. We would be happy to include CRPS alongside our existing metrics in the camera-ready version. We originally chose CWR because it directly captures the trade-off between coverage (PICP) and interval width (NMPIW) in an interpretable manner, but we acknowledge that CRPS could offer additional insights into the quality of our predictions.
>
> > [W4] Interpretation of results and conclusions
>
> We acknowledge the reviewer's concern about result interpretation. The performance decline occurs for the missing data results because our current empirical missing data simulator ($p_\xi$) may not fully capture the underlying data distribution. This highlights a significant trade-off: While SEMF provides a model-agnostic framework for handling missing data, its performance depends on the quality of the missing data simulator. We observe better results with complete data because the framework can directly learn from actual data distributions rather than simulated ones.
>
> > [W5] Running time
>
> We thank the reviewer for this suggestion. We agree that reporting computational costs would provide valuable context; however, as pointed out in Section 2, we are training the same type of model at each iteration of EM. Hence, the computational cost is naturally significantly higher. With that said, we firmly believe there are good strategies to reduce the overall number of iterations needed for training. We would gladly report the current training times, but they will unsurprisingly be much higher than the baselines yet feasible enough to run on a local machine.
>
> We thank the reviewer again for their insightful comments.

---

### Note · Authors · 2024-11-26

**Comment:**

We thank all the reviewers for their feedback on our paper. After careful consideration, we have decided to withdraw from ICLR. Once again, we thank the reviewers for their time

**Withdrawal Confirmation:**

I have read and agree with the venue's withdrawal policy on behalf of myself and my co-authors.